# AN EMPIRICAL STUDY OF PRE-TRAINED VISION MODELS ON OUT-OF-DISTRIBUTION GENERALIZATION

## ABSTRACT

Generalizing to out-of-distribution (OOD) data – that is, data from domains unseen during training – is a key challenge in modern machine learning, which has only recently received much attention. Some existing approaches propose leveraging larger models and pre-training on larger datasets. In this paper, we provide new insights in applying these approaches. Concretely, we show that larger models and larger datasets need to be *simultaneously* leveraged to improve OOD performance on image classification. Moreover, we show that using smaller learning rates during fine-tuning is critical to achieving good results, contrary to popular intuition that larger learning rates generalize better when training from scratch. We show that strategies that improve in-distribution accuracy may, counter-intuitively, lead to poor OOD performance despite strong in-distribution performance. Our insights culminate to a method that achieves state-of-the-art results on a number of OOD generalization benchmark tasks, often by a significant margin.

## 1 INTRODUCTION

Most machine learning (ML) models assume that test data is drawn from the same distribution as training data. However, this assumption does not hold in many real-world applications. As a result, ML models often fail to generalize to out-of-distribution (OOD) data encountered during their deployment and suffer from significant performance drops compared with the model performance on in-distribution (ID) data (Quiñonero-Candela et al., 2009; Torralba & Efros, 2011). For example, common distribution shifts prevalent during test time include variation in locations (Koh et al., 2021) and weather (Volk et al., 2019), noise and blur corruptions (Hendrycks & Dietterich, 2018), and small adversarial perturbations (Szegedy et al., 2013). As ML models are increasingly deployed in safety-critical applications, it is becoming ever more critical to ensure strong *OOD generalization* for such models, i.e., the models robustly generalizing to relevant OOD data not seen during training.

While this problem is indeed difficult since the goal is to generalize to data that are not seen during training, there have been a handful of methods recently proposed to improve OOD generalization. Some methods propose specialized training methods, such as simulating OOD data during training (Li et al., 2018a), learning invariant representations (Arjovsky et al., 2019), and performing adversarial data augmentation (Volpi et al., 2018). Intriguingly, Gulrajani & Lopez-Paz (2020) conducted an extensive empirical evaluation on domain generalization benchmark datasets, and demonstrated that classical empirical risk minimization (ERM) approach achieves nearly state-of-the-art OOD generalization performance compared with these specialized methods. On the other hand, most of the approaches apply small or medium size networks that are usually pre-trained on the ImageNet-1k dataset, such as pre-trained ResNet50 (Gulrajani & Lopez-Paz, 2020). On the other hand, recent works find that pre-training on larger and more diverse data is one of the most effective paths toward generalizing to out-of-distribution data on ImageNet (Taori et al., 2020).

In this paper, we systematically investigate the importance of pre-trained models for OOD generalization. Specifically, we conduct extensive experiments on models with different model sizes that are pre-trained on large datasets. The pre-trained models are then fine-tuned on the training data for the underlying task. Instead of focusing on achieving state-of-the-art results on benchmark OOD datasets, our empirical study aims to develop a better understanding of the critical role that pre-trained models along with different design choices for fine-tuning such models play in ensuring

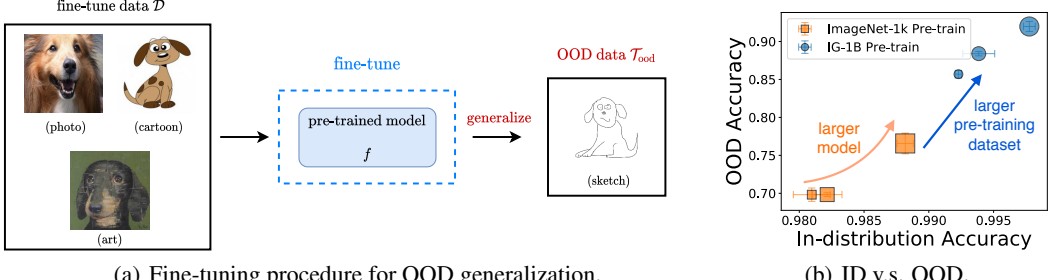

(a) Fine-tuning procedure for OOD generalization.      (b) ID v.s. OOD.

Figure 1: (**Left**) Illustration of our fine-tuning procedure used throughout this paper, i.e., we fine-tune a pre-trained model on training dataset $\mathcal{D}$ and evaluate the model performance on OOD data $\mathcal{T}_{\text{ood}}$ − a data domain not seen during training. (**Right**) Evaluating ID and OOD accuracies for two classes of pre-trained models. Orange squares represent the ImageNet-1k pre-trained models and blue circles represent the IG-1B pre-trained models, where IG-1B is a larger dataset than ImageNet-1k. Larger marker size means the model size is larger.

good OOD generalization. This provides novel insights towards closing the gap between ID and OOD generalization for future research.

The main contributions of our work are as follows:

- When leveraging larger models for OOD generalization, we find that both large pre-training dataset sizes and large model sizes are critical. Missing either one of the two components may hurt OOD generalization.

- We show that using a small learning rate generalizes better for OOD when we leverage a pre-trained model. This is a complementary argument to Li et al. (2019) that suggested to use a large learning rate for better generalization in the case of no pre-training.

- We show cases where improving in-distribution performance actually leads to worse OOD performance suggesting that ID performance is not a reliable indicator of OOD performance.

- Our insights culminate in a method that achieves SOTA, often by a significant margin, on a number of OOD generalization benchmark tasks including PACS, VLCS, and Office-Home.

## 2    PRELIMINARIES AND EXPERIMENTAL SETUP

We begin this section by introducing the *out-of-distribution generalization* problem, with the primary focus on a special case of this broader issue, namely *domain generalization*. Subsequently, we describe the experimental setup adopted in our empirical study and various parameter choices we make throughout the paper.

In this paper, we study a general multi-class classification setting, with all input instances and their labels belonging to $\mathcal{X}$ and $\mathcal{Y} := \{1, \ldots, K\}$, respectively. Let $\mathcal{D}$ denote the training dataset which may potentially comprises data belong to $k$ different training domains $\{\mathcal{D}^j\}_{j \in [k] := \{1, \ldots, k\}}$, i.e., $\mathcal{D} = \cup_{j \in [k]} \mathcal{D}^j$. We assume that the training data from the $j$-th domain $\mathcal{D}^j = \{(\boldsymbol{x}_i^j, y_i^j)\}_{i \in [n_j]} \subset \mathcal{X} \times \mathcal{Y}$ is sampled from the distribution $P_{\text{id}}^j$, i.e., $(\boldsymbol{x}_i^j, y_i^j) \sim P_{\text{id}}^j$. At test time, we evaluate a trained model for both its in-distribution and out-of-distribution performance. The in-distribution performance is evaluated on a test dataset $\mathcal{T}_{\text{id}}$ that consists of instances belonging to the domains encountered in the training dataset $\mathcal{D}$. On the other hand, we utilize an OOD test dataset $\mathcal{T}_{\text{ood}}$ from an unseen domain with distribution $P_{\text{ood}}$ to assess the model's OOD performance.

Given a family of candidate classifiers $\mathcal{F} = \{f : \mathcal{X} \to \mathcal{Y}\}$ and the underlying training dataset $\mathcal{D}$, we primarily employ standard *empirical risk minimization* (ERM) (Vapnik, 1998) to learn a classifier $\hat{f}$ as follows:

$$\hat{f} = \underset{f \in \mathcal{F}}{\operatorname{argmin}} \ \frac{1}{|\mathcal{D}|} \sum_{i \in \mathcal{D}} \ell(f(\boldsymbol{x}_i), y_i), \qquad (1)$$

where $\ell(\cdot, \cdot)$ denotes the cross-entropy loss function. For a classifier $f$, we define its *in-distribution accuracy* $\texttt{Acc}_{\mathrm{id},f}$ and *out-of-distribution accuracy* $\texttt{Acc}_{\mathrm{ood},f}$ as follows:

$$\texttt{Acc}_{\mathrm{id},f} = \mathbb{E}_{(\boldsymbol{x},y)\in\mathcal{T}_{\mathrm{id}}}\big[\mathbb{1}\big\{\widehat{f}(\boldsymbol{x}) \neq y\big\}\big]; \quad \texttt{Acc}_{\mathrm{ood},f} = \mathbb{E}_{(\boldsymbol{x},y)\in\mathcal{T}_{\mathrm{ood}}}\big[\mathbb{1}\big\{\widehat{f}(\boldsymbol{x}) \neq y\big\}\big], \tag{2}$$

where $\mathbb{1}\{\cdot\}$ denotes the standard indicator function. Often, models that achieve large in-distribution accuracy $\texttt{Acc}_{\mathrm{id}}$ only achieve relatively small out-of-distribution accuracy $\texttt{Acc}_{\mathrm{ood}}$, i.e., $\texttt{Acc}_{\mathrm{id}} \gg \texttt{Acc}_{\mathrm{ood}}$ (Torralba & Efros, 2011; Hendrycks & Gimpel, 2016; Gulrajani & Lopez-Paz, 2020). Thus, under the domain generalization problem, one particularly focuses on designing training methods that result in classifier with good performance on both out-of-distribution data and in-distribution data.

**Pre-trained models.** We mainly focus on fine-tuning pre-trained models on the training dataset $\mathcal{D}$ by using ERM. We explore four classes of pre-trained models for OOD generalization: (1). ResNet-based models (He et al., 2016a; Xie et al., 2017) pre-trained on ImageNet (Russakovsky et al., 2015) (ResNet50, ResNext50-32x4d, and ResNext101-32x8d); (2). (BiTm)-ResNet-v2-based models (He et al., 2016b) pre-trained on ImageNet-21k (Deng et al., 2009) (ResNetV2-50x1, ResNetV2-50x3, and ResNetV2-101x1) (Kolesnikov et al., 2020), where group normalization (Wu & He, 2018) and weight standardization (Qiao et al., 2019) are used in ResNetV2; (3). (SWSL)-ResNet-based semi-weakly supervised ImageNet models pre-trained on IG-1B-Targeted data (Yalniz et al., 2019); and (4). Vision transformer (ViT) pre-trained on ImageNet-21k (Dosovitskiy et al., 2020). A detailed description of these pre-trained models can be found in Table 3 (in Appendix).

**OOD datasets.** In our experiments, we use four vision datasets used to benchmark domain generalization algorithms (Gulrajani & Lopez-Paz, 2020): (1). **PACS** dataset (Li et al., 2017); (2). **Office-Home** dataset (Venkateswara et al., 2017); (3). **VLCS** dataset (Fang et al., 2013); and (4). **TerraIncognita** dataset (Beery et al., 2018). Each of these datasets contains 4 different domains. We train the models on 3 domains and treat the examples from the remaining domain as the out-of-distribution data $\mathcal{T}_{\mathrm{ood}}$. For the three training domains, we use $80\%$ data as training dataset and the remaining $20\%$ data for evaluation. We add the test domain information after the dataset to specify the test domain, for example, **PACS** (S) means the training domains are 'P', 'A', and 'C' and the test (OOD) domain is 'S'.

**Fine-tuning and model selection.** We use stochastic gradient descent (SGD) with a momentum of 0.9 for fine-tuning all pre-trained models considered in this paper. The default weight decay for SGD is set to be 0. We use a cosine learning rate decay (Loshchilov & Hutter, 2016) as learning rate scheduler for SGD. For initial learning rates $\eta$ we compare the following set $\{0.05, 0.02, 0.01, 0.005, 0.002, 0.001, 0.0005, 0.0002, 0.0001\}$. For evaluation, we pick the five checkpoints from each model with highest in-distribution accuracy. We then compute the average OOD accuracy of these five checkpoints.

## 3 MAIN RESULTS

We present our main experimental results in this section. First, we highlight the importance of models pre-trained on large and diverse datasets for OOD generalization. Next, we investigate the effect of fine-tuning learning rates, especially for models pre-trained on diverse datasets. Then, we perform a systematic examination of several components of pre-trained models for OOD generalization, including model size, pre-training dataset, and model architecture. Finally, we evaluate whether techniques used for improving ID accuracy can also enhance OOD generalization.

### 3.1 IMPORTANCE OF A BETTER PRE-TRAINED MODEL

Models pre-trained on more diverse datasets have been shown to achieve better OOD generalization on real-world distribution shifts (Taori et al., 2020; Hendrycks et al., 2020a). As a warm-up for understanding the properties of pre-trained models on OOD data, we first study the OOD generalization performance of a ResNet-based model that is pre-trained on a large and diverse pre-training dataset. In particular, we focus on an SWSL-ResNext101-32x4d model pre-trained on the IG-1B-Targeted data (Yalniz et al., 2019), which is a much larger and more diverse dataset than ImageNet.

The fine-tuning approach described in Section 2 with different learning rate significantly outperforms the baseline results from Gulrajani & Lopez-Paz (2020) (cf. Table 1). This shows that a better pre-trained model indeed improves OOD generalization without using specialized algorithms for domain

Table 1: Comparison with ERM baseline results from Gulrajani & Lopez-Paz (2020). We compare our approach with the baseline in terms of the OOD accuracy $\text{Acc}_{\text{ood}}$. Note that our approach amounts to fine-tuning SWSL-ResNext101-32x4d (Yalniz et al., 2019) with different learning rates and employing the models selection procedure described in Section 2. Note that, for each benchmark, we treat one of the four domains as the OOD domain and fine-tune the model on the remaining three domains. We report results for all four choices for the OOD domain on each benchmark.

| | **OOD Domain** | | | | | **OOD Domain** | | | |
|---|---|---|---|---|---|---|---|---|---|
| **PACS** | A | C | P | S | **VLCS** | C | L | S | V |
| Baseline | 88.1 | 78.0 | 97.8 | 79.1 | Baseline | 97.6 | 63.3 | 72.2 | 76.4 |
| Ours | 96.2 | 94.6 | 99.4 | 91.3 | Ours | 98.2 | 66.1 | 77.0 | 80.5 |
| **Office-Home** | A | C | P | R | **TerraIncognita** | L100 | L38 | L43 | L46 |
| Baseline | 62.7 | 53.4 | 76.5 | 77.3 | Baseline | 50.8 | 42.5 | 57.9 | 37.6 |
| Ours | 76.4 | 68.5 | 86.5 | 87.6 | Ours | 48.3 | 47.5 | 57.2 | 43.7 |

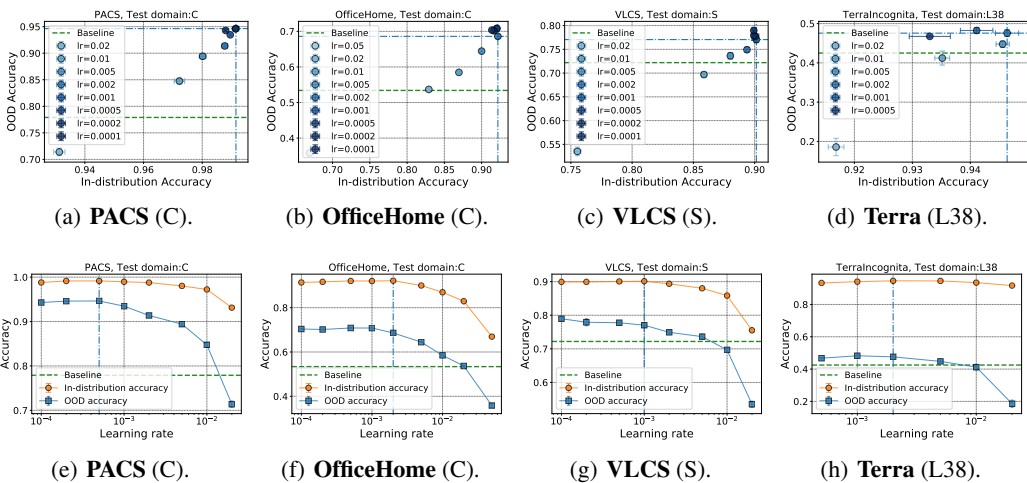

(a) **PACS** (C).      (b) **OfficeHome** (C).      (c) **VLCS** (S).      (d) **Terra** (L38).

(e) **PACS** (C).      (f) **OfficeHome** (C).      (g) **VLCS** (S).      (h) **Terra** (L38).

Figure 2: Evaluating models (SWSL-ResNext101-32x4d) fine-tuned using different learning rates on ID and OOD data. (Top row) Scatter plot of ID accuracy ($X$-axis) and OOD accuracy ($Y$-axis). (Bottom row) Compare ID accuracy with OOD accuracy w.r.t. learning rate ($X$-axis). The green dashed line corresponds to the baseline OOD accuracy, and the blue dash-dotted line represents the selected model (by selecting the model with best ID accuracy).

generalization. For example, the OOD accuracy improves from $79.1\%$ to $91.3\%$ on **PACS** (S), and $62.7\%$ to $76.4\%$ on **Office-Home** (A). Overall, Table 1 suggests that using a larger model pre-trained on more data can be very effective for better OOD generalization. Next, we conduct more detailed experiments to better understand the OOD generalization performance of various pre-trained models.

## 3.2 EFFECT OF FINE-TUNE LEARNING RATE

Given the fact that simply fine-tuning the SWSL model leads to compelling improvement in the OOD generalization, we now take a closer look at models trained with different learning rates. We evaluate models trained with different learning rates on both ID and OOD test data. Figure 2 summarizes the results for fine-tuning SWSL-ResNext101-32x4d with different learning rates. In Figure 2(a)-2(d), each point in the plot corresponds to a model trained with a distinct learning rate. As mentioned in Section 2, we select models that achieves the top-5 ID accuracy, and depict the standard deviation of both $\text{Acc}_{\text{id}}$ and $\text{Acc}_{\text{ood}}$ for each model. We only show results for models that achieve $> 95\%$ *training* accuracy to better compare model performance.

**Regularization effect of small learning rate.** Based on Figure 2, our main finding is that the fine-tuning learning rate plays a key role in determining both ID and OOD accuracy. In particular, we

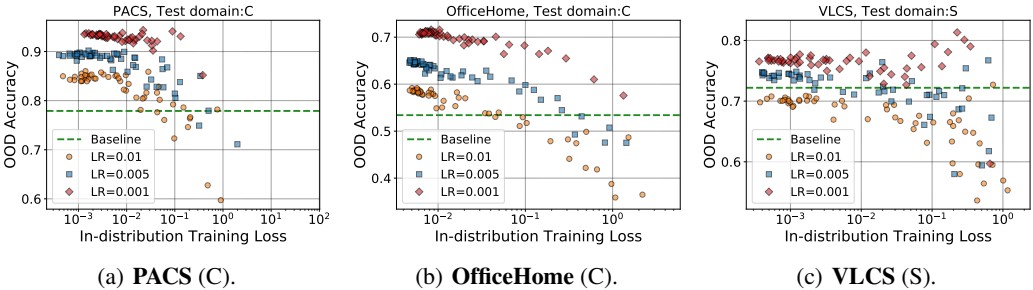

(a) **PACS** (C).      (b) **OfficeHome** (C).      (c) **VLCS** (S).

Figure 3: OOD accuracy of models (SWSL-ResNext101-32x4d) during training. We visualize models trained with three different learning rates in terms of OOD accuracy vs. training loss. Each point in the above plots represents the model evaluated at one iteration during training

observe that, in most of the settings considered in this paper, the models trained with smaller learning rates achieve much better OOD generalization[1], even when the ID accuracy does not change much. This is different from the fact that larger learning rates generalize better when the model is directly trained on the training data $\mathcal{D}$ without a pre-training phase (Li et al., 2019; Lewkowycz et al., 2020; Nakkiran, 2020). We also observe similar behavior with SWSL-ResNext101-32x8d models, where smaller learning rates lead to better OOD performance (see Figure 9 in Appendix). Intriguingly, we do not observe this phenomenon for models pre-trained on ImageNet. There, as shown in Figure 12 in Appendix, when the ID accuracy is similar, a model trained with larger learning rate achieves better OOD generalization (e.g., see Figure 12(e) and 12(f)). To summarize, our results indicates that when models are pre-trained on more diverse datasets, smaller learning rates can lead to better OOD generalization.

To obtain a better understanding of the effect of learning rate, we also study the OOD accuracy during training. In Figure 3, we visualize the OOD accuracy vs. training loss as measured every 100 SGD iterations for models trained with three different learning rates $\eta \in \{0.01, 0.005, 0.001\}$. Note that all three learning rates eventually achieve similar training loss, but the models trained with smaller learning rates have better OOD generalization. Figure 3 also suggests that models trained with larger learning rates cannot achieve similar OOD accuracy by using early stopping. Meanwhile, the figure also confirms the regularization effect of small learning rates for fine-tuning pre-trained models.

Overall, we find that the learning rate is a key parameter for achieving good OOD generalization. While different learning rates may not affect ID accuracy much, OOD accuracy can be very sensitive to the choice of learning rate. Our results also highlight a limitation of performing model selection based on ID accuracy as models with similar ID accuracy may have very different OOD performance.

### 3.3 PRE-TRAINING FOR BETTER OOD GENERALIZATION

Now, we systematically explore the role that pre-trained models play in improving OOD generalization. Specifically, we study three aspects of pre-trained models: pre-training dataset, model architecture, and model size. Furthermore, we compare the models trained from scratch (i.e., without pre-training) with pre-trained models with respect to OOD generalization.

**Effect of pre-training dataset.** We study the OOD performance of models pre-trained on different datasets, inlcuding ResNext101-32x8d pre-trained on ImageNet, BiTm-ResNetV2-50x3 pretrained on ImageNet-21k, and SWSL-ResNext101-32x8d pre-trained on IG-1B-Targeted. The SWSL model and BiTm model achieve similar Top-1 accuracies on ImageNet, $84.2\%$ and $84.0\%$, respectively, and the standard ResNext101-32x8d achieves a slightly worse in terms of Top-1 accuracy of $79.3\%$. The results for this comparison are summarized in Figure 4 and Figure 15 (in Appendix).

Our first observation is that the pre-training dataset has a big impact on OOD generalization performance. With same architecture and model size, the SWSL pre-trained model consistently outperforms the standard ImageNet pre-trained model across. Secondly, we find that SWSL generally performs the best among all three models as it is pre-trained on the largest and the most diverse pre-training dataset. Furthermore, we find that the BiTm model pre-trained on ImageNet-21k also generally

---

[1]Among models achieve similar ID generalization, models trained with smaller learning rates achieve better OOD generalization.

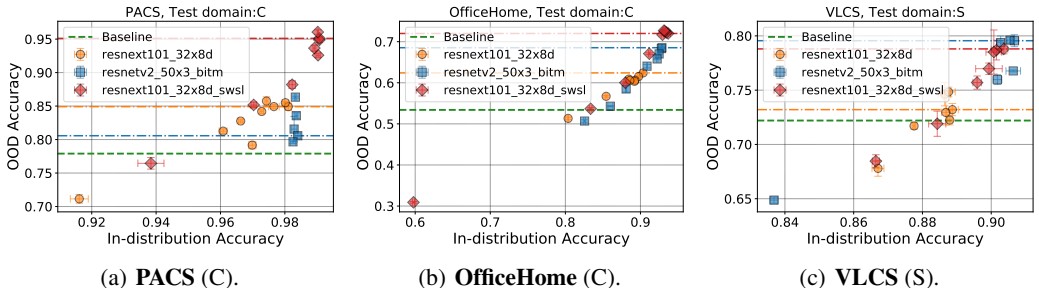

(a) **PACS** (C).    (b) **OfficeHome** (C).    (c) **VLCS** (S).

Figure 4: Evaluating OOD and ID performance of models pre-trained on different datasets. Each color corresponds to the models pre-trained on a distinct dataset and the dash-dotted line represents the model picked by our model selection procedure. For each model we report the accuracy across different fine-tuning learning rates.

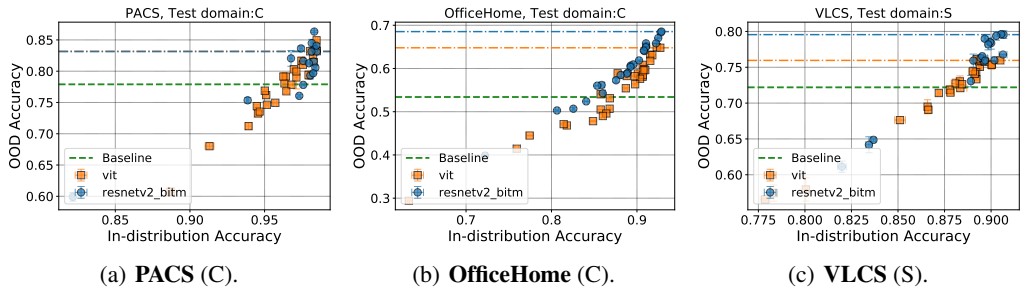

(a) **PACS** (C).    (b) **OfficeHome** (C).    (c) **VLCS** (S).

Figure 5: A comparison of four ViT models and three BiTm models on OOD accuracy and ID accuracy. The orange squares represent ViT models and the blue circles represent BiTm models. The dash-dotted lines represent the selected models. We do not distinguish the model architectures within the same model class.

performs better than the ResNext model pre-trained on ImageNet (except for the PACS dataset, where both models have similar OOD performance).

Another interesting finding is that the OOD accuracy can be significantly improved by chosing the right fine-tuning learning rate, while the improvement in the ID accuracy is small. For example, in Figure 4(a), the ID accuracy improves from 98% to 99%, whereas the OOD accuracy increases $\sim 10\%$. Our results suggest that the "*accuracy on the line*" phenomenon (Miller et al., 2021) does not always hold for the benchmark datasets used in domain generalization. This also echoes the observation from D'Amour et al. (2020), that OOD generalization can be very different among models with similar ID accuracies.

**Effect of model architecture: ViTs vs. CNNs.** We now investigate the role that the model architecture plays in ensuring good OOD generalization. Compared with convolutional neural networks (CNNs), the recently proposed Vision Transformers (ViT) achieves similar or even better performance on image classification tasks (Dosovitskiy et al., 2020). This raises the question whether vision transformers behave differently from CNNs in terms of their OOD performance? Towards this, we explore three BiT-ResNetV2 pre-trained models (Kolesnikov et al., 2020) and four ViT pre-trained models (Dosovitskiy et al., 2020). In particular, we compare BiTm-ResNetV2-{50x1, 101x1, 50x3} with ViT-{small-patch32, small-patch16, base-patch32, base-patch16}. We present the comparison between ViTs and BiTms on OOD benchmarks in Figure 5 and Figure 16 (in Appendix).

We find that the OOD generalization accuracy of ViT models is similar to that of BiTm models. In some cases, BiTs slightly outperform ViTs on OOD generalization, for example, results shown in Figure 5(b), 5(c), 16(a). Since both classes of models are pre-trained on the same pre-training dataset (i.e., ImageNet-21k) and achieve similar ImageNet Top-1 accuracies (84.5% for ViTs vs. 84.0% for BiTms), our results indicate that replacing convolution operation with self-attention operation does not bring additional benefits in terms of OOD generalization for the settings we consider in this paper.

**Effect of model size.** It is evident from Figure 4, where both the baseline model (i.e., ResNet50) and ResNext101-32x8d are pre-trained on the ImageNet-1k dataset, that increasing the model

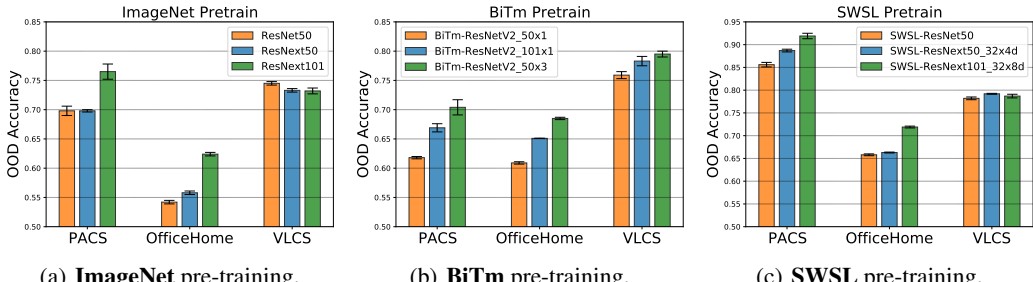

(a) **ImageNet** pre-training.   (b) **BiTm** pre-training.   (c) **SWSL** pre-training.

Figure 6: Evaluating OOD generalization for three classes of models with different model sizes. **Left:** Results for ResNe(x)t models pre-trained on ImageNet-1k. **Middle:** Results for BiTm-ResNetV2 models pre-trained on ImageNet-21k. **Right:** Results for SWSL-ResNe(x)t models pre-trained on IG-1B-Targeted. For a given model class, each model size is represented by a distinct color.

size from ResNet50 to ResNext101-32x8d can improve OOD generalization. For example, OOD generalization is improved from 78.0% to 84.9% in Figure 4(a). Motivated by these promising results, we conduct experiments to understand the role of model size for OOD generalization. We consider increasing model size on three classes of models, including ResNe(x)t pre-trained on ImageNet-1k, BiTm-ResNetV2 pre-trained on ImageNet-21k, and ResNe(x)t pre-trained on IG-1B-Targeted. We investigate three model sizes for each class of pre-trained models and the results are summarized in Figure 6 and Table 4-6 (in Appendix).

For all three classes of models, we find that increasing the model size can improve OOD generalization in many settings. Overall, this shows that model size is a crucial design choice for improving OOD generalization. Furthermore, we note that SWSL-ResNet50 is pre-trained on a larger and more diverse dataset than BiTm-ResNetV2-50x3 and both models achieve $\sim 100\%$ training accuracy, but SWSL-ResNet50 has lower OOD accuracy than BiTm-ResNetV2-50x3 on **OfficeHome** (C). This suggests that *both* the pre-training dataset and the model size play key role in determining the OOD generalization, and ideally it is preferable to employ larger models pre-trained on larger and more diverse pre-training data.

**OOD performance of models trained from random initialization.**    To further validate the importance of pre-training data along with the model size for better OOD generalization, we train models with increasing model sizes from random initialization, i.e., without employing a pre-training phase. Our results for this experiment (cf. Table 8) show that larger models do not significantly improve the OOD generalization without the use of pre-training. For example, increasing ResNext50-32x4d to ResNext101-32x8d only improves the OOD accuracy on **OfficeHome** (C) by less than 2%. Thus, our results suggest that without pre-training, only increasing model size is not very effective in improving the OOD performance.

### 3.4   ANALYSIS OF TECHNIQUES USED FOR IMPROVING ID ACCURACY

In this subsection, we examine various techniques that have been shown to improve ID accuracy to assess their utility in achieving good OOD generalization. First, we study the impact of training data size on the OOD generalization, as increasing the number of training samples is an effective approach to improve ID generalization. Then we evaluate four techniques in our OOD setting, including label smoothing (Szegedy et al., 2016), AutoAugment (Cubuk et al., 2018), PatchGaussian (Lopes et al., 2019), and Sharpness-Aware Minimization (SAM) (Foret et al., 2020).

**Utility of more training data.**    We consider fine-tuning with datsets of four different sizes, i.e., $100\%, 50\%, 25\%,$ and $12.5\%$ of the total training data for a given benchmark. Our results in Figure 7 show that increasing the training data from $12.5\%$ to $100\%$ does not significantly improve OOD generalization. In fact, on **VLCS** (S), a better OOD generalization is realized when we utilize less training data to train a ResNext101-32x8d model (cf. Figure 7(c)). This suggests that increasing ID training samples is not as effective as using larger and better pre-trained models.

**Methods for improving ID accuracy.**    We find that applying the methods listed in Table 2 does not significantly improve the OOD generalization across four datasets compared with scaling model size and pre-training dataset size, and utilizing augmentations/regularization can potentially even hurt

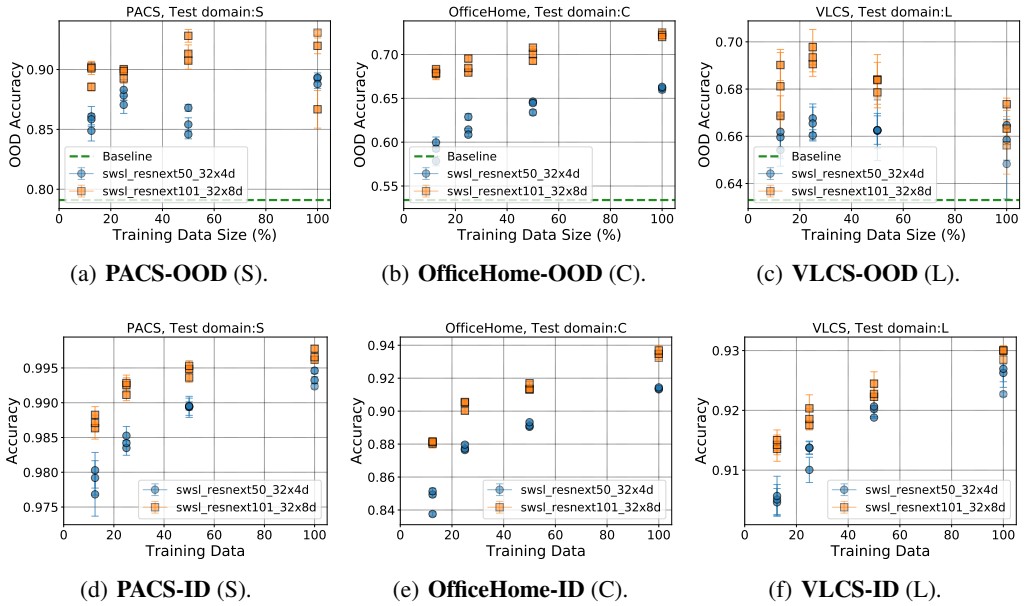

Figure 7: Evaluating OOD generalization performance of models trained with different number of training samples. $X$-axis represents the number of training samples. We use SWSL-ResNext50-32x4d and SWSL-ResNext101-32x8d as the pre-trained models. For each pre-trained model, we visualize the OOD accuracies of the top-3 models selected by ID accuracy.

Table 2: Evaluation of four techniques (label smoothing, AutoAugment, PatchGaussian, and SAM) for OOD generalization. We use the same pre-trained model (SWSL-ResNext101-32x4d) across all settings. The number inside the parentheses after the method name represents the value of the technique-specific hyperparameter, e.g., PatchGaussian (1.0) corresponds to employing PatchGaussian (Lopes et al., 2019) with $\sigma = 1.0$. We highlight the best two OOD accuracies for each dataset with bold text.

| Method | PACS (C) | Office (C) | VLCS (L) | Terra (L46) |
|---|---|---|---|---|
| ERM (in Table 1) | **94.6** | 68.5 | 66.1 | 43.7 |
| Label Smoothing (0.1) | 91.6 | 70.6 | 65.2 | 42.5 |
| Label Smoothing (0.2) | 93.5 | 70.8 | 66.3 | **44.6** |
| AutoAugment | 93.5 | 70.7 | 65.2 | 38.1 |
| PatchGaussian (1.0) | 92.8 | 65.8 | 64.6 | 13.1 |
| PatchGaussian (0.5) | **94.4** | 69.3 | 63.6 | 9.7 |
| SAM (0.02) | 93.5 | **72.2** | **67.1** | **44.7** |
| SAM (0.05) | 92.8 | **71.3** | **67.5** | 42.3 |

OOD generalization. For example, applying PatchGaussian decreases OOD accuracy on **Terra** (L46) from 43.7% to 13.1% and 9.7% with $\sigma = 1.0$ and 0.5, respectively. In constrast, PatchGaussian has very minimal impact on the ID performance, where it achieves $\text{Acc}_{\text{id}} = 95.8\%$ and $\text{Acc}_{\text{id}} = 95.6\%$, respectively; ERM attains $\text{Acc}_{\text{id}} = 95.9\%$. Contrary to our results, Lopes et al. (2019) notice that PatchGaussian can improve both the clean (ID) accuracy and robustness to common corruptions (OOD accuracy). This suggests that one should employing augmentation/regularization techniques carefully so as to not harm OOD generalization as a side effect. On the other hand, SAM with parameter[2] $\rho = 0.02$ improves the OOD generalization on three benchmarks. Overall, we find that methods used for improving ID accuracy do not necessarily improve OOD accuracy, when compared with the simple ERM-based fine-tuning approach.

---

[2]The perturbation parameter $\rho$ is defined in Foret et al. (2020).

## 4 RELATED WORK

**Domain adaptation and domain generalization.** Ben-David et al. (2010) proposed $\mathcal{H}\Delta\mathcal{H}$-divergence and applied it to develop error bounds on new test data distribution that are different from the training distribution. Motivated by the theoretical results in Ben-David et al. (2010), a large body of work was devoted to learning domain-invariant representations for domain adaptation where unlabeled data from the target domain are available during training (Ajakan et al., 2014; Ganin et al., 2016; Tzeng et al., 2017; Zhao et al., 2018). Different from the domain adaptation problem, no information from the target domain is available in the domain generalization problem. Recent works proposed various algorithms to improve domain generalization, including learning invariant representations across training domains (Li et al., 2018b; Arjovsky et al., 2019), distributionally robust optimization Sagawa et al. (2020), data augmentation (Volpi et al., 2018; Zhou et al., 2020), and causal framework (Heinze-Deml & Meinshausen, 2017; Mahajan et al., 2021). Zhou et al. (2021); Wang et al. (2021) provide a more detailed review of the topic of domain generalization.

**OOD generalization and model robustness.** Recent works developed new datasets to evaluate model robustness to out-of-distribution data, including ImageNet-V2 (Recht et al., 2019), CIFAR-10.1 (Recht et al., 2018), ImageNet-C and CIFAR-10-C (Hendrycks & Dietterich, 2018), ImageNet-R (Hendrycks et al., 2020a), WILDS (Koh et al., 2021), etc. A line of work proposed methods to improve model robustness to commons corruptions (Lopes et al., 2019; Xie et al., 2020; Hendrycks et al., 2020b; Calian et al., 2021; Yi et al., 2021). Xie et al. (2021) investigated how to leverage auxiliary information and pre-training to improve OOD generalization. On multiple datasets, the researchers have observed a near linear correlation between the OOD accuracy and the ID accuracy (Recht et al., 2019; Miller et al., 2020; Mania & Sra, 2020; Miller et al., 2021). However, Taori et al. (2020) found that most approaches, including the ones that improve robustness to synthetic distribution shifts, do not improve model robustness to natural distribution shifts, except for the models that are pre-trained on larger datasets. Similar observation has been made in Hendrycks et al. (2020a). Our work focuses on the domain generalization benchmark datasets as well as how to perform fine-tuning one various pre-trained models for better OOD generalization.

## 5 DISCUSSION AND FUTURE WORK

Pre-training is one of the most effective approaches for improving model performance in a wide range of machine learning tasks. In this work, we perform an empirical study of fine-tuning a diverse set of pre-trained models and evaluate their OOD generalization. We find that simply fine-tuning *larger* models pre-trained on *more (diverse) data* can significantly improve model performance on OOD data. Additionally, and we also identify the *regularization effect of small learning rates* that is important for achieving better OOD generalization. Further, through extensive experimentation, we demonstrate that, while relying on pre-trained models, *model size* and *pre-training dataset* play a key role in ensuring good OOD generalization. We hope our results can further inspire future research on bridging the gap between in-distribution accuracy and out-of-distribution accuracy. There are multiple interesting immediate directions to explore in future work that we discuss next.

*Scaling model size and pre-training dataset size.* Our empirical results indicate that model size and the pre-training dataset size are two essential factors for improving OOD performance. Even though we primarily focus on image classification benchmarks, similar behavior has been observed in NLP domain (Brown et al., 2020), where increasing model size leads to monotonic improvements in zero-shot performance on unseen tasks for example. It is worth noting that while increasing model size or pre-training dataset size only marginally improves the in-distribution accuracy, the increase in OOD accuracy can be much larger. It is an interesting direction to explore the differences in ID and OOD generalization in other domains.

*How to perform model selection?* Our study indicates that models trained via different methods can exhibit a large variation in terms of their OOD performance, even when their in-distribution accuracy is very similar, e.g., see Figure 4(c) and Figure 5(c). We thus believe that the development of better model selection approaches for OOD generalization will be a key direction of future work.

*Limitations of ERM.* Despite impressive results for OOD generalization, we find that ERM-based fine-tuning of pre-trained models is unlikely to close the in-distribution to out-of-distribution generalization gap , even when the model size or the pre-training dataset becomes much larger. To generalize to unseen domains, it might be necessary to have access to extra information about the OOD dataset (e.g., unlabeled OOD data as available in the domain adaptation setting (Pan & Yang, 2009)) and/or design novel algorithms (e.g., updating the model parameters during test time (Sun et al., 2020)).

**Reproducibility Statement.**    In Section 2 and Appendix A.2, We provide detailed descriptions on the implementations used in this paper, including about the pre-trained models, datasets, models selection, hyperparameters, and how to train the models.

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

# A APPENDIX

## A.1 OTHER RELATED WORKS

**Pre-training and fine-tuning.** Li et al. (2020) examined how to select hyperparameters for fine-tuning on a various benchmark vision datasets. Dodge et al. (2020) performed an extensive study on fine-tuning pre-trained language models on NLP benchmarks, and investigated the impact of weight initialization and training data order on model performance. Kaplan et al. (2020) studied the scaling laws, including pre-training dataset size, model size, and compute budget for training, for language models. They found that increasing model size for pre-training can improve model performance on OOD data. More recently, Zhai et al. (2021) explored how to improve ViT model performance on ImageNet and few-shot learning tasks through scaling the model size and data size.

## A.2 ADDITIONAL EXPERIMENTAL DETAILS

**Implementation details.** Our implementation is mainly adapted from **DomainBed** (developed in Gulrajani & Lopez-Paz (2020)). Before we fine-tune the pre-trained models, we replace the final linear layer of the pre-trained model with a random initialized linear layer with output dimension equals to the number of classes for the dataset. Most of the pre-trained models are from PyTorch **Im**age **M**odels (**timm**). The default training iteration is $5,000$. For the training from scratch setting (i.e., without pre-training), we set the training iteration as $10,000$ and weight decay as $10^{-5}$. During the fine-tuning, we save model checkpoint every $100$ iterations and select five checkpoints that achieves the top-5 in-distribution accuracies. The input size of the ViT models is $(3, 224, 224)$. In Table 3, we summarize the top-1/top-5 ImageNet accuracies of pre-trained models used in this paper.

Table 3: Top-1 accuracy and Top-5 accuracy of pre-trained models considered in this paper. (**ImageNet**) ResNet-based models pre-trained on ImageNet-1k. (**BiTm**) ResNetV2-based models pre-trained on ImageNet-21k. (**SWSL**) ResNet-based models pre-trained on IG-1B-Targeted. (**ViT**) Vision transformer-based models pre-trained on ImageNet-21k.

| **ImageNet** | ResNet50 | ResNext50-32x4d | ResNext101-32x8d | |
|---|---|---|---|---|
| Top-1 Acc | 76.13 | 77.62 | 79.30 | |
| Top-5 Acc | 92.86 | 93.69 | 94.51 | |
| **BiTm** | ResNetV2-50x1 | ResNetV2-101x1 | ResNetV2-50x3 | |
| Top-1 Acc | 80.34 | 82.33 | 84.01 | |
| Top-5 Acc | 95.68 | 96.51 | 97.12 | |
| **SWSL** | ResNet50 | ResNext50-32x4d | ResNext101-32x4d | ResNext101-32x8d |
| Top-1 Acc | 81.16 | 82.18 | 83.23 | 84.28 |
| Top-5 Acc | 95.97 | 96.23 | 96.76 | 97.17 |
| **ViT** | Small-Patch32 | Base-Patch32 | Small-Patch16 | Base-Patch16 |
| Top-1 Acc | 75.99 | 80.72 | 81.40 | 84.53 |
| Top-5 Acc | 93.27 | 95.56 | 96.13 | 97.29 |

## A.3 ADDITIONAL EXPERIMENTAL RESULTS

In this subsection, we present additional experimental results in Section 3.

**Effect of fine-tune learning rate in Section 3.2.** In addition to Figure 2 and Figure 3, we provide more experimental results on the effect of fine-tune learning rate. For fine-tuning results with different learning rates, we present more results for fine-tuning SWSL-ResNext101-32x4d in Figure 8, results for fine-tuning SWSL-ResNext101-32x8d in Figure 9, results for fine-tuning SWSL-ResNext50-32x4d in Figure 10, results for fine-tuning BiTm-ResNetV2-50x3 in Figure 11, results for fine-tuning ResNext101-32x8d (ImageNet pre-trained) in Figure 12, and results for fine-tuning ResNext50-32x4d (ImageNet pre-trained) in Figure 13. Meanwhile, in Figure A.3, we provide more results on visualizing the OOD accuracy vs. training loss for SWSL-ResNext101-32x4d.

**Effect of pre-training dataset.**    In Figure 15, we provide additional experimental results on studying the effect of pre-training dataset.

**Effect of model architecture: ViTs vs. CNNs.**    In Figure 16, we provide additional experimental results on comparing the performance of ViTs and CNNs.

**Effect of model size.**    We provide additional results on comparing models with different model sizes in Table 4 (ImageNet-1k pre-trained models), Table 5 (BiTm pre-trained models (Kolesnikov et al., 2020)), Table 6 (SWSL pre-trained models (Yalniz et al., 2019)), Table 7 (ViT pre-trained models (Dosovitskiy et al., 2020)), and Table 8 (without pre-training).

**Utility of more training data.**    In Figure 17, we provide additional experimental results on investigating the impact of the number of training samples on the **TerraIncognita** dataset, including the ID/OOD accuracy results.

**Methods for improving ID accuracy.**    In Table 9, we provide the in-distribution accuracy evaluations of the methods described in Table 2. Also, in Table 10 and 11, we provide additional results on the ID/OOD accuracy evaluations of different methods (on more datasets).

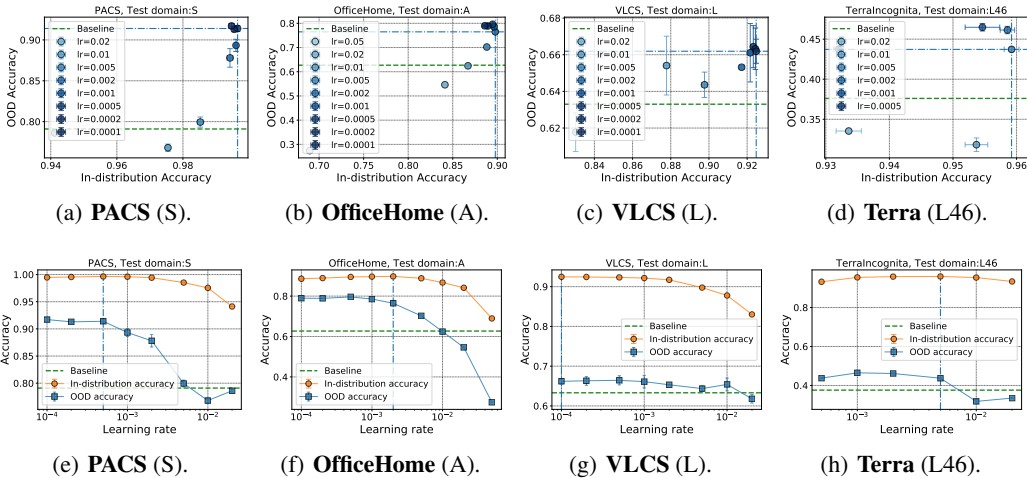

Figure 8: (Additional results) Evaluating models (SWSL-ResNext101-32x4d) fine-tuned by different learning rates on in-distribution and out-of-distribution data. (Top row) Scatter plot of in-distribution accuracy ($X$-axis) and out-of-distribution accuracy ($Y$-axis). (Bottom row) Compare in-distribution accuracy with out-of-distribution accuracy w.r.t. learning rate ($X$-axis). The green dashed line corresponds to the baseline OOD accuracy, and the blue dash-dotted line represents the selected model (by selecting the model with best in-distribution accuracy).

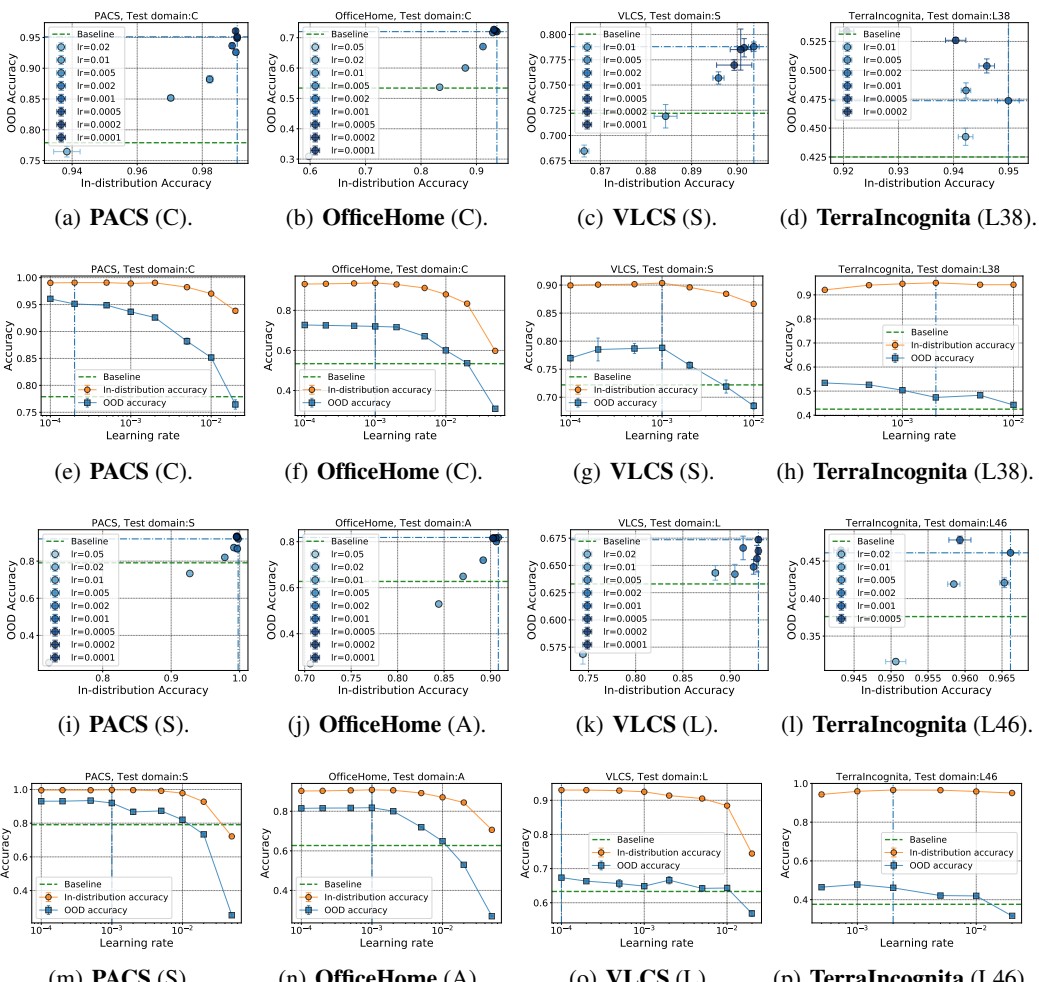

Figure 9: Evaluating models (SWSL-ResNext101-32x8d) fine-tuned by different learning rates on in-distribution and out-of-distribution data. (Top row) Scatter plot of in-distribution accuracy ($X$-axis) and out-of-distribution accuracy ($Y$-axis). (Bottom row) Compare in-distribution accuracy with out-of-distribution accuracy w.r.t. learning rate ($X$-axis). The green dashed line corresponds to the baseline OOD accuracy, and the blue dash-dot line represents the selected model (by selecting the model with best in-distribution accuracy).

Table 4: Comparing ImageNet pre-trained models with different model sizes. We evaluate both in-distribution accuracy and out-of-distribution. (*left*: ID accuracy, *right*: OOD accuracy).

| Models | **PACS** (C) | **PACS** (S) |
|---|---|---|
| ResNet50 | $0.977 \pm 0.000 / 0.785 \pm 0.006$ | $0.980 \pm 0.001 / 0.698 \pm 0.008$ |
| ResNext50-32x4d | $0.977 \pm 0.001 / 0.825 \pm 0.005$ | $0.982 \pm 0.001 / 0.698 \pm 0.002$ |
| ResNext101-32x8d | $0.981 \pm 0.000 / 0.849 \pm 0.005$ | $0.988 \pm 0.000 / 0.765 \pm 0.013$ |

| Models | **Office-Home** (A) | **Office-Home** (C) |
|---|---|---|
| ResNet50 | $0.869 \pm 0.001 / 0.662 \pm 0.002$ | $0.874 \pm 0.002 / 0.542 \pm 0.003$ |
| ResNext50-32x4d | $0.874 \pm 0.000 / 0.645 \pm 0.003$ | $0.887 \pm 0.002 / 0.558 \pm 0.003$ |
| ResNext101-32x8d | $0.887 \pm 0.000 / 0.717 \pm 0.002$ | $0.903 \pm 0.000 / 0.624 \pm 0.003$ |

| Models | **VLCS** (L) | **VLCS** (S) |
|---|---|---|
| ResNet50 | $0.898 \pm 0.000 / 0.645 \pm 0.003$ | $0.885 \pm 0.001 / 0.745 \pm 0.003$ |
| ResNext50-32x4d | $0.905 \pm 0.000 / 0.646 \pm 0.007$ | $0.891 \pm 0.002 / 0.733 \pm 0.003$ |
| ResNext101-32x8d | $0.909 \pm 0.001 / 0.643 \pm 0.002$ | $0.888 \pm 0.001 / 0.732 \pm 0.005$ |

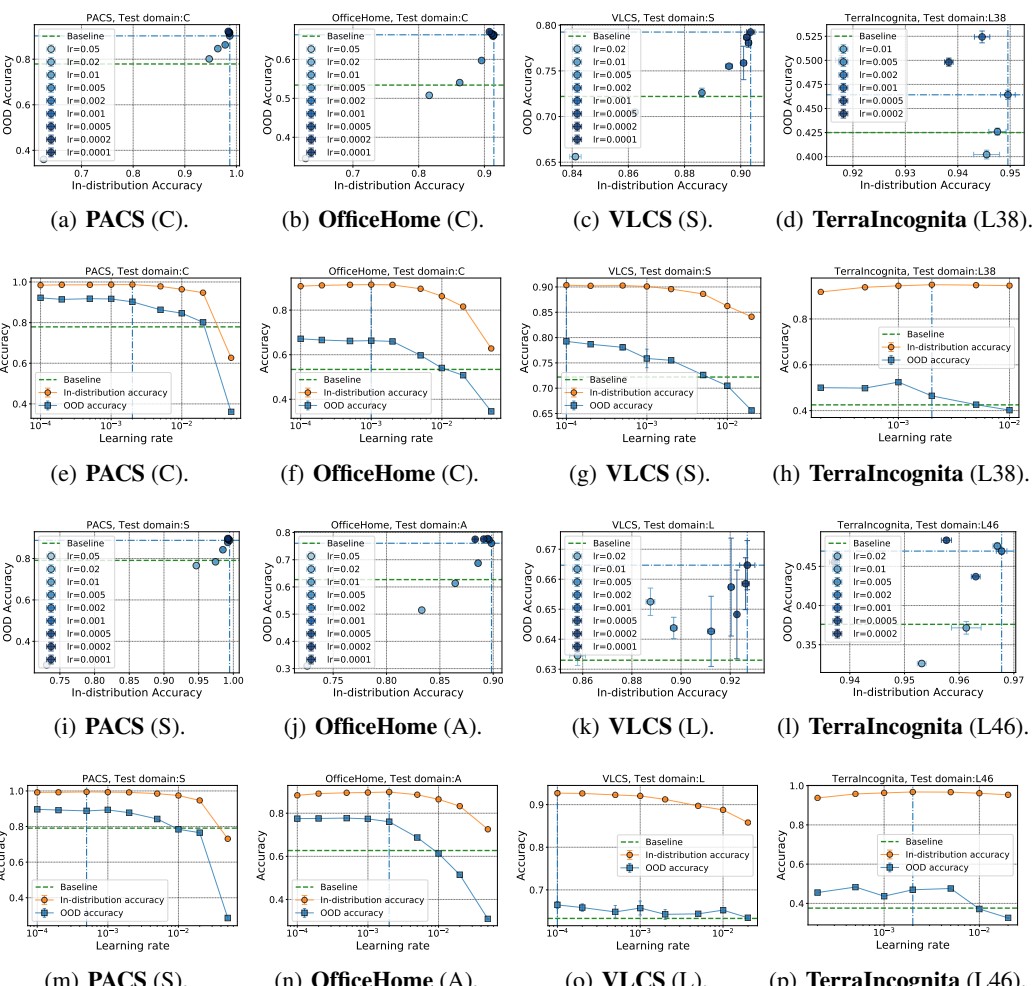

(a) **PACS** (C).  (b) **OfficeHome** (C).  (c) **VLCS** (S).  (d) **TerraIncognita** (L38).

(e) **PACS** (C).  (f) **OfficeHome** (C).  (g) **VLCS** (S).  (h) **TerraIncognita** (L38).

(i) **PACS** (S).  (j) **OfficeHome** (A).  (k) **VLCS** (L).  (l) **TerraIncognita** (L46).

(m) **PACS** (S).  (n) **OfficeHome** (A).  (o) **VLCS** (L).  (p) **TerraIncognita** (L46).

Figure 10: Evaluating models (SWSL-ResNext50-32x4d) fine-tuned by different learning rates on in-distribution and out-of-distribution data. (Top row) Scatter plot of in-distribution accuracy ($X$-axis) and out-of-distribution accuracy ($Y$-axis). (Bottom row) Compare in-distribution accuracy with out-of-distribution accuracy w.r.t. learning rate ($X$-axis). The green dashed line corresponds to the baseline OOD accuracy, and the blue dash-dot line represents the selected model (by selecting the model with best in-distribution accuracy).

Table 5: Comparing BiTm models with different model sizes. We evaluate both in-distribution accuracy and out-of-distribution. (*left*: ID accuracy, *right*: OOD accuracy).

| Models | **PACS** (C) | **PACS** (S) |
|---|---|---|
| BiTm-ResNetV2-50x1 | $0.980 \pm 0.001$ / $0.793 \pm 0.005$ | $0.983 \pm 0.001$ / $0.618 \pm 0.002$ |
| BiTm-ResNetV2-101x1 | $0.984 \pm 0.000$ / $0.831 \pm 0.002$ | $0.990 \pm 0.001$ / $0.669 \pm 0.007$ |
| BiTm-ResNetV2-50x3 | $0.984 \pm 0.000$ / $0.805 \pm 0.004$ | $0.989 \pm 0.000$ / $0.704 \pm 0.013$ |

| Models | **Office-Home** (A) | **Office-Home** (C) |
|---|---|---|
| BiTm-ResNetV2-50x1 | $0.887 \pm 0.001$ / $0.721 \pm 0.001$ | $0.895 \pm 0.001$ / $0.609 \pm 0.002$ |
| BiTm-ResNetV2-101x1 | $0.904 \pm 0.001$ / $0.756 \pm 0.001$ | $0.910 \pm 0.000$ / $0.651 \pm 0.000$ |
| BiTm-ResNetV2-50x3 | $0.912 \pm 0.000$ / $0.792 \pm 0.001$ | $0.928 \pm 0.000$ / $0.685 \pm 0.002$ |

| Models | **VLCS** (L) | **VLCS** (S) |
|---|---|---|
| BiTm-ResNetV2-50x1 | $0.922 \pm 0.000$ / $0.647 \pm 0.010$ | $0.896 \pm 0.000$ / $0.759 \pm 0.006$ |
| BiTm-ResNetV2-101x1 | $0.923 \pm 0.002$ / $0.656 \pm 0.001$ | $0.900 \pm 0.000$ / $0.783 \pm 0.008$ |
| BiTm-ResNetV2-50x3 | $0.931 \pm 0.001$ / $0.655 \pm 0.009$ | $0.906 \pm 0.001$ / $0.795 \pm 0.005$ |

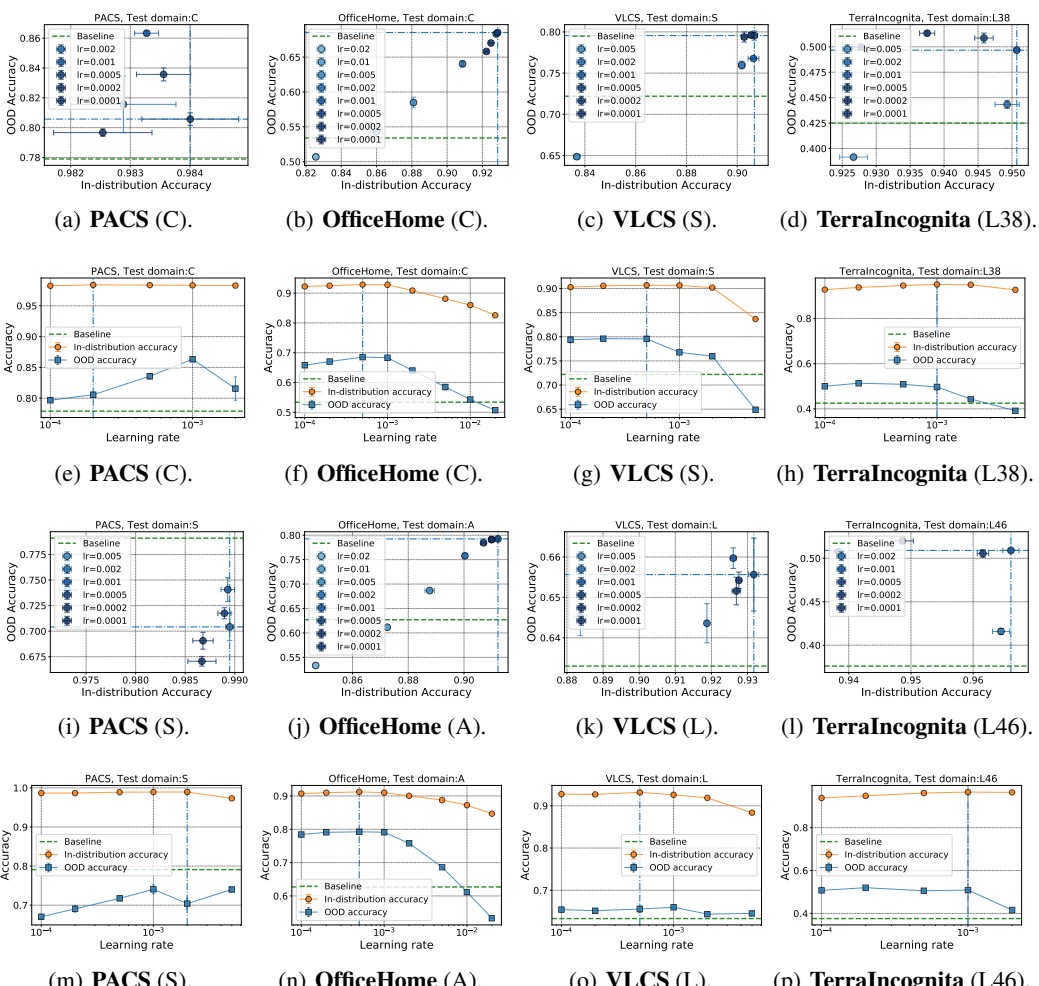

Figure 11: Evaluating BiTm models (BiTm-ResNetV2-50x3) fine-tuned by different learning rates on in-distribution and out-of-distribution data. (Top row) Scatter plot of in-distribution accuracy ($X$-axis) and out-of-distribution accuracy ($Y$-axis). (Bottom row) Compare in-distribution accuracy with out-of-distribution accuracy w.r.t. learning rate ($X$-axis). The green dashed line corresponds to the baseline OOD accuracy, and the blue dash-dot line represents the selected model (by selecting the model with best in-distribution accuracy).

Table 6: Comparing SWSL models with different model sizes. We evaluate both in-distribution accuracy and out-of-distribution. (*left*: ID accuracy, *right*: OOD accuracy).

| Models | **PACS** (C) | **PACS** (S) |
|---|---|---|
| SWSL-ResNet50 | $0.987 \pm 0.000 / 0.870 \pm 0.008$ | $0.992 \pm 0.000 / 0.856 \pm 0.005$ |
| SWSL-ResNext50-32x4d | $0.987 \pm 0.000 / 0.902 \pm 0.002$ | $0.994 \pm 0.000 / 0.887 \pm 0.003$ |
| SWSL-ResNext101-32x8d | $0.990 \pm 0.000 / 0.951 \pm 0.002$ | $0.997 \pm 0.000 / 0.919 \pm 0.006$ |

| Models | **Office-Home** (A) | **Office-Home** (C) |
|---|---|---|
| SWSL-ResNet50 | $0.886 \pm 0.000 / 0.729 \pm 0.003$ | $0.908 \pm 0.001 / 0.658 \pm 0.002$ |
| SWSL-ResNext50-32x4d | $0.898 \pm 0.000 / 0.760 \pm 0.001$ | $0.914 \pm 0.001 / 0.663 \pm 0.001$ |
| SWSL-ResNext101-32x8d | $0.908 \pm 0.000 / 0.818 \pm 0.001$ | $0.936 \pm 0.002 / 0.719 \pm 0.002$ |

| Models | **VLCS** (L) | **VLCS** (S) |
|---|---|---|
| SWSL-ResNet50 | $0.920 \pm 0.000 / 0.662 \pm 0.006$ | $0.900 \pm 0.000 / 0.782 \pm 0.003$ |
| SWSL-ResNext50-32x4d | $0.926 \pm 0.003 / 0.664 \pm 0.008$ | $0.903 \pm 0.001 / 0.792 \pm 0.001$ |
| SWSL-ResNext101-32x8d | $0.930 \pm 0.000 / 0.673 \pm 0.002$ | $0.903 \pm 0.001 / 0.787 \pm 0.004$ |

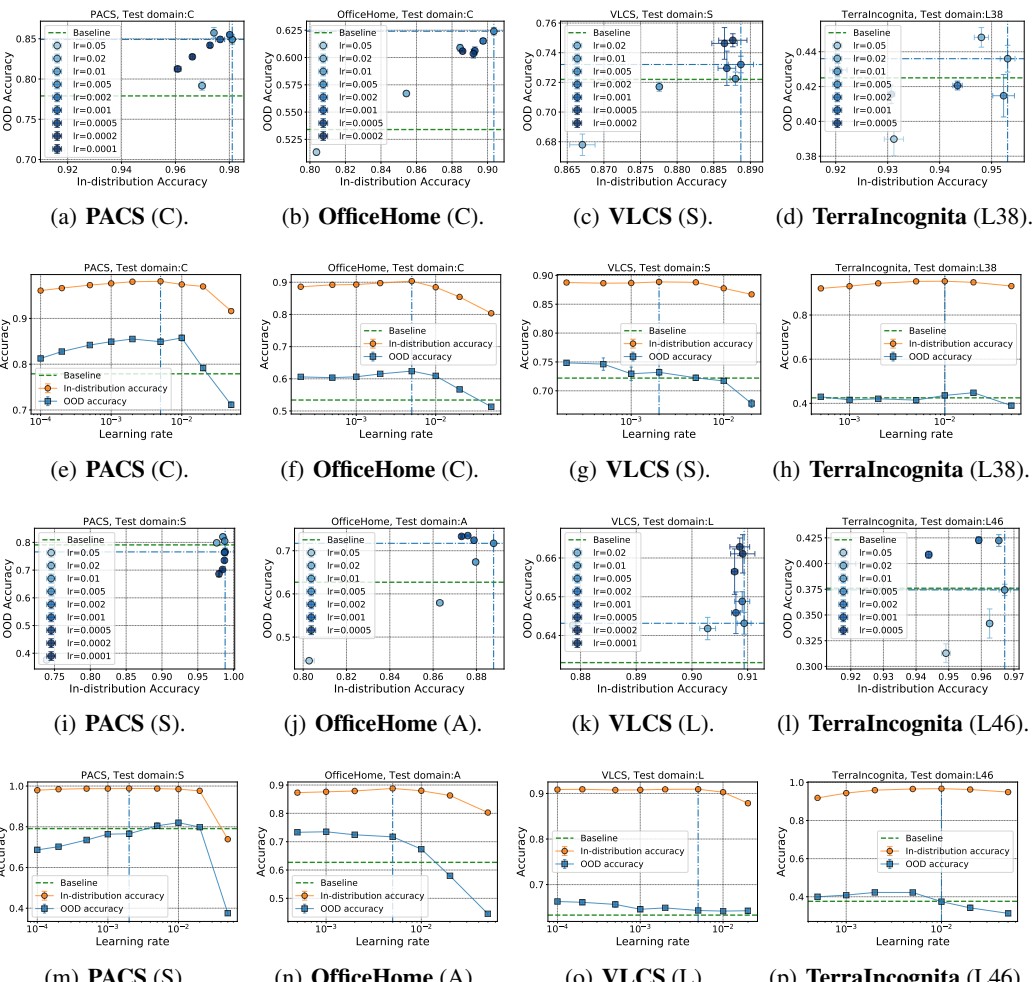

Figure 12: Evaluating models (ResNext101-32x8d pre-trained on ImageNet) fine-tuned by different learning rates on in-distribution and out-of-distribution data. (Top row) Scatter plot of in-distribution accuracy ($X$-axis) and out-of-distribution accuracy ($Y$-axis). (Bottom row) Compare in-distribution accuracy with out-of-distribution accuracy w.r.t. learning rate ($X$-axis). The green dashed line corresponds to the baseline OOD accuracy, and the blue dash-dot line represents the selected model (by selecting the model with best in-distribution accuracy).

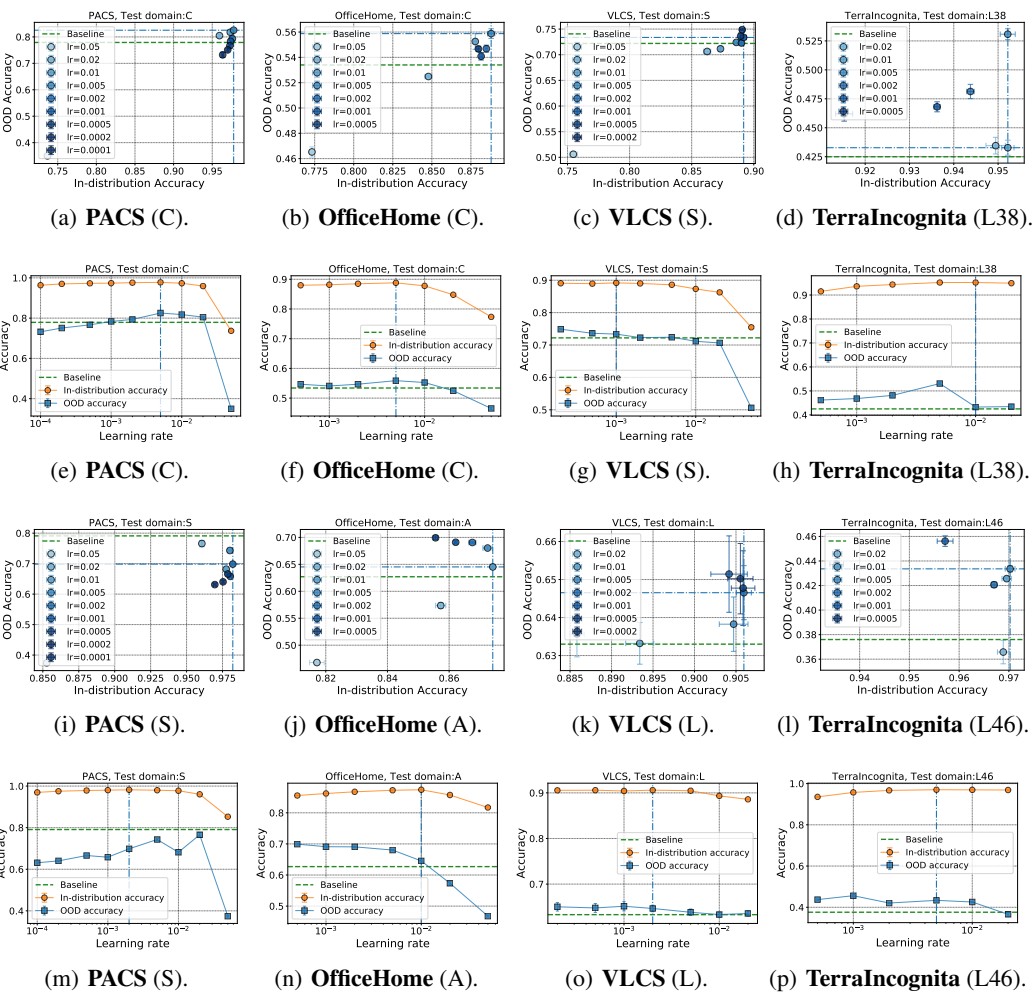

Figure 13: Evaluating models (ResNext50-32x4d pre-trained on ImageNet) fine-tuned by different learning rates on in-distribution and out-of-distribution data. (Top row) Scatter plot of in-distribution accuracy ($X$-axis) and out-of-distribution accuracy ($Y$-axis). (Bottom row) Compare in-distribution accuracy with out-of-distribution accuracy w.r.t. learning rate ($X$-axis). The green dashed line corresponds to the baseline OOD accuracy, and the blue dash-dot line represents the selected model (by selecting the model with best in-distribution accuracy).

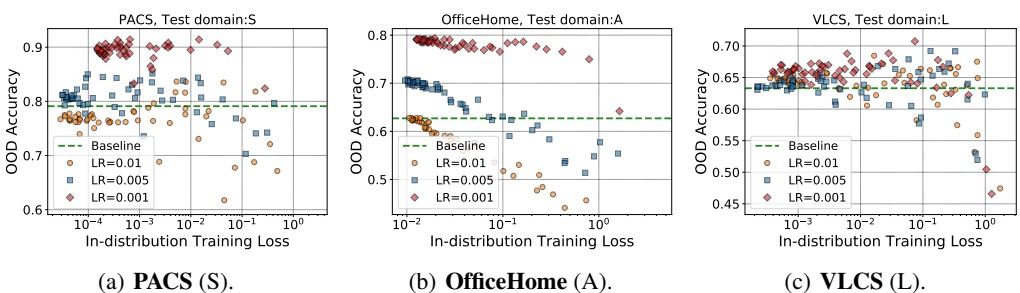

Figure 14: OOD accuracy of models (SWSL-ResNext101-32x4d) during training. We visualize models trained with three different learning rates in terms of OOD accuracy v.s. training loss. Each point in the above plots represents the model evaluated at one iteration during training.

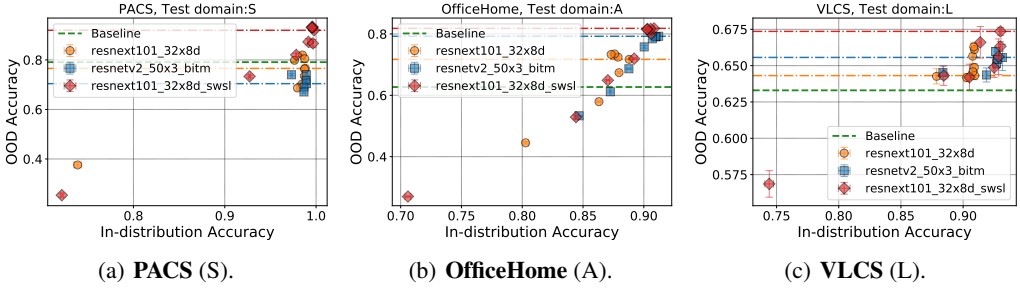

(a) **PACS** (S).  (b) **OfficeHome** (A).  (c) **VLCS** (L).

Figure 15: Evaluating out-of-distribution and in-distribution performance of models pre-trained on different datasets. Each color corresponds to models pre-trained on one dataset and the dash-dot line represents the selected model.

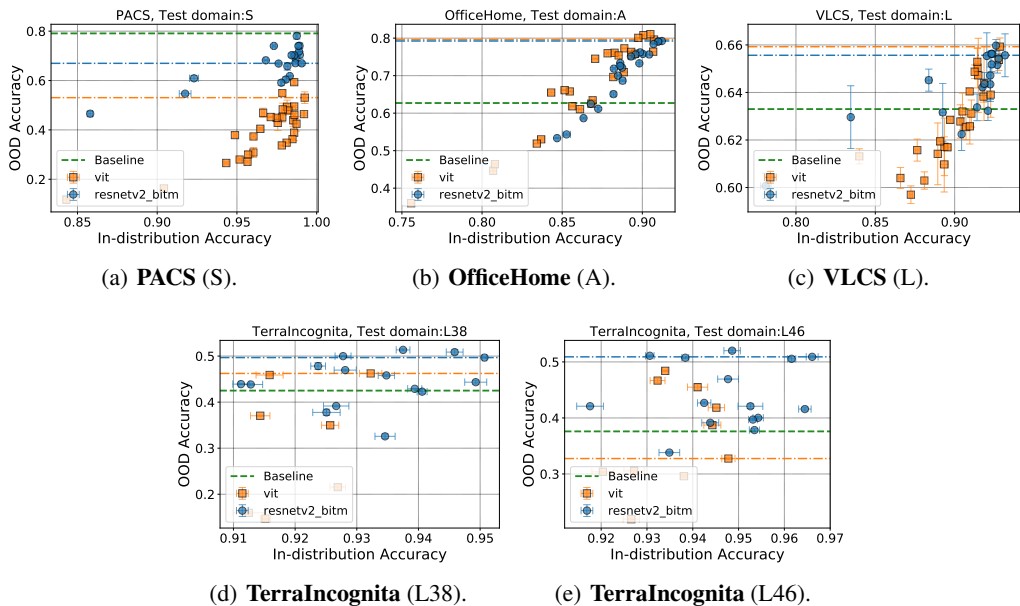

(a) **PACS** (S).  (b) **OfficeHome** (A).  (c) **VLCS** (L).

(d) **TerraIncognita** (L38).  (e) **TerraIncognita** (L46).

Figure 16: A comparison of four ViT models and three BiTm models on out-of-distribution accuracy and in-distribution accuracy. The orange squares represent ViT models and the blue circles represent BiTm models. The dash-dot lines represent the selected models. We do not distinguish the model architectures within the same model class.

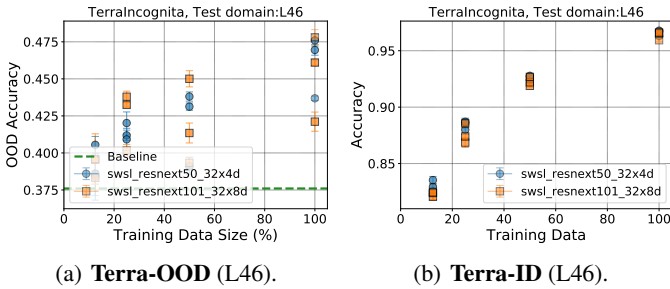

(a) **Terra-OOD** (L46).  (b) **Terra-ID** (L46).

Figure 17: Evaluating OOD generalization performance of models trained with different number of training samples on the **TerraIncognita** dataset. $X$-axis represents the number of training samples. We use SWSL-ResNext50-32x4d and SWSL-ResNext101-32x8d as the pre-trained models. For each pre-trained model, we visualize the OOD accuracies of the top-3 models selected by ID accuracy.

Table 7: Comparing ViTs models with different model sizes. We evaluate both in-distribution accuracy and out-of-distribution. (*left*: ID accuracy, *right*: OOD accuracy).

| Models | **PACS** (C) | **PACS** (S) |
|---|---|---|
| ViT-small-patch32 | $0.951 \pm 0.001$ / $0.746 \pm 0.001$ | $0.964 \pm 0.001$ / $0.404 \pm 0.005$ |
| ViT-small-patch16 | $0.982 \pm 0.000$ / $0.825 \pm 0.005$ | $0.985 \pm 0.000$ / $0.494 \pm 0.004$ |
| ViT-base-patch32 | $0.971 \pm 0.000$ / $0.789 \pm 0.018$ | $0.982 \pm 0.000$ / $0.478 \pm 0.013$ |
| ViT-base-patch16 | $0.985 \pm 0.000$ / $0.832 \pm 0.001$ | $0.992 \pm 0.000$ / $0.530 \pm 0.023$ |
| Models | **Office-Home** (A) | **Office-Home** (C) |
| ViT-small-patch32 | $0.856 \pm 0.001$ / $0.618 \pm 0.001$ | $0.867 \pm 0.000$ / $0.506 \pm 0.001$ |
| ViT-small-patch16 | $0.888 \pm 0.001$ / $0.772 \pm 0.001$ | $0.907 \pm 0.001$ / $0.584 \pm 0.003$ |
| ViT-base-patch32 | $0.895 \pm 0.001$ / $0.758 \pm 0.004$ | $0.909 \pm 0.001$ / $0.596 \pm 0.001$ |
| ViT-base-patch16 | $0.907 \pm 0.000$ / $0.796 \pm 0.001$ | $0.927 \pm 0.001$ / $0.647 \pm 0.001$ |
| Models | **VLCS** (L) | **VLCS** (S) |
| ViT-small-patch32 | $0.897 \pm 0.001$ / $0.628 \pm 0.001$ | $0.883 \pm 0.003$ / $0.721 \pm 0.004$ |
| ViT-small-patch16 | $0.918 \pm 0.001$ / $0.643 \pm 0.015$ | $0.894 \pm 0.001$ / $0.759 \pm 0.004$ |
| ViT-base-patch32 | $0.910 \pm 0.001$ / $0.631 \pm 0.005$ | $0.892 \pm 0.002$ / $0.733 \pm 0.001$ |
| ViT-base-patch16 | $0.928 \pm 0.001$ / $0.659 \pm 0.003$ | $0.904 \pm 0.000$ / $0.759 \pm 0.000$ |

Table 8: Comparing models trained from scratch with different model sizes. We evaluate both in-distribution accuracy and out-of-distribution. (*left*: ID accuracy, *right*: OOD accuracy).

| Models | **PACS** (C) | **PACS** (S) |
|---|---|---|
| ResNet50 | $0.812 \pm 0.003$ / $0.470 \pm 0.005$ | $0.817 \pm 0.005$ / $0.294 \pm 0.004$ |
| ResNext50-32x4d | $0.816 \pm 0.003$ / $0.497 \pm 0.004$ | $0.785 \pm 0.001$ / $0.192 \pm 0.011$ |
| ResNext101-32x8d | $0.681 \pm 0.004$ / $0.505 \pm 0.010$ | $0.770 \pm 0.002$ / $0.287 \pm 0.007$ |
| Models | **Office-Home** (A) | **Office-Home** (S) |
| ResNet50 | $0.643 \pm 0.002$ / $0.201 \pm 0.001$ | $0.557 \pm 0.004$ / $0.226 \pm 0.004$ |
| ResNext50-32x4d | $0.656 \pm 0.000$ / $0.225 \pm 0.001$ | $0.560 \pm 0.002$ / $0.233 \pm 0.001$ |
| ResNext101-32x8d | $0.654 \pm 0.002$ / $0.216 \pm 0.001$ | $0.575 \pm 0.001$ / $0.249 \pm 0.001$ |
| Models | **VLCS** (L) | **VLCS** (S) |
| ResNet50 | $0.758 \pm 0.003$ / $0.570 \pm 0.012$ | $0.747 \pm 0.001$ / $0.509 \pm 0.000$ |
| ResNext50-32x4d | $0.757 \pm 0.000$ / $0.567 \pm 0.013$ | $0.751 \pm 0.001$ / $0.513 \pm 0.005$ |
| ResNext101-32x8d | $0.760 \pm 0.002$ / $0.567 \pm 0.004$ | $0.739 \pm 0.003$ / $0.502 \pm 0.002$ |

Table 9: *In-discribution* accuracy evaluation of four techniques (label smoothing, AutoAugment, PatchGaussian, and SAM) for OOD generalization. We use the same pre-trained model (SWSL-ResNext101-32x4d) across all settings. The number inside the parentheses after the method name represents the value of the technique-specific hyperparameter, e.g., PatchGaussian (1.0) corresponds to employing PatchGaussian (Lopes et al., 2019) with $\sigma = 1.0$.

| Method | **PACS** (C) | **Office** (C) | **VLCS** (L) | **Terra** (L46) |
|---|---|---|---|---|
| ERM (in Table 1) | 99.1 | 92.1 | 92.4 | 95.9 |
| Label Smoothing (0.1) | 99.0 | 92.4 | 92.9 | 95.8 |
| Label Smoothing (0.2) | 99.0 | 91.8 | 93.1 | 95.8 |
| AutoAugment | 98.4 | 90.6 | 91.9 | 93.8 |
| PatchGaussian (1.0) | 99.1 | 92.9 | 92.6 | 95.8 |
| PatchGaussian (0.5) | 99.0 | 92.3 | 92.6 | 95.6 |
| SAM (0.02) | 99.2 | 92.9 | 93.3 | 96.1 |
| SAM (0.05) | 99.0 | 93.2 | 93.0 | 95.1 |

Table 10: *Out-of-discrimination* accuracy evaluation of four techniques (label smoothing, AutoAugment, PatchGaussian, and SAM) for OOD generalization. We use the same pre-trained model (SWSL-ResNext101-32x4d) across all settings. The number inside the parentheses after the method name represents the value of the technique-specific hyperparameter, e.g., PatchGaussian (1.0) corresponds to employing PatchGaussian (Lopes et al., 2019) with $\sigma = 1.0$. We highlight the best two OOD accuracies for each dataset with bold text.

| Method | **PACS** (S) | **Office** (A) | **VLCS** (S) | **Terra** (L38) |
|---|---|---|---|---|
| ERM (in Table 1) | 91.3 | 76.4 | 77.0 | 47.5 |
| Label Smoothing (0.1) | **91.7** | 78.0 | 78.0 | **48.3** |
| Label Smoothing (0.2) | **91.7** | 78.6 | 78.6 | **50.3** |
| AutoAugment | 91.3 | 78.1 | 77.1 | 46.9 |
| PatchGaussian (1.0) | 90.3 | 66.4 | 71.6 | 12.4 |
| PatchGaussian (0.5) | 90.0 | 73.5 | 75.9 | 6.2 |
| SAM (0.02) | 89.7 | **80.3** | **79.2** | 42.0 |
| SAM (0.05) | 90.7 | **80.4** | **79.7** | 43.9 |

Table 11: *In-discrimination* accuracy evaluation of four techniques (label smoothing, AutoAugment, PatchGaussian, and SAM) for OOD generalization. We use the same pre-trained model (SWSL-ResNext101-32x4d) across all settings. The number inside the parentheses after the method name represents the value of the technique-specific hyperparameter, e.g., PatchGaussian (1.0) corresponds to employing PatchGaussian (Lopes et al., 2019) with $\sigma = 1.0$.

| Method | **PACS** (S) | **Office** (A) | **VLCS** (S) | **Terra** (L38) |
|---|---|---|---|---|
| ERM (in Table 1) | 99.6 | 89.7 | 90.1 | 94.6 |
| Label Smoothing (0.1) | 99.5 | 89.8 | 90.6 | 94.8 |
| Label Smoothing (0.2) | 99.5 | 89.8 | 90.5 | 94.5 |
| AutoAugment | 99.6 | 88.6 | 89.9 | 92.2 |
| PatchGaussian (1.0) | 99.6 | 90.1 | 90.1 | 94.0 |
| PatchGaussian (0.5) | 99.6 | 90.2 | 90.3 | 94.1 |
| SAM (0.02) | 99.2 | 91.0 | 90.9 | 94.5 |
| SAM (0.05) | 99.7 | 91.2 | 90.9 | 93.3 |

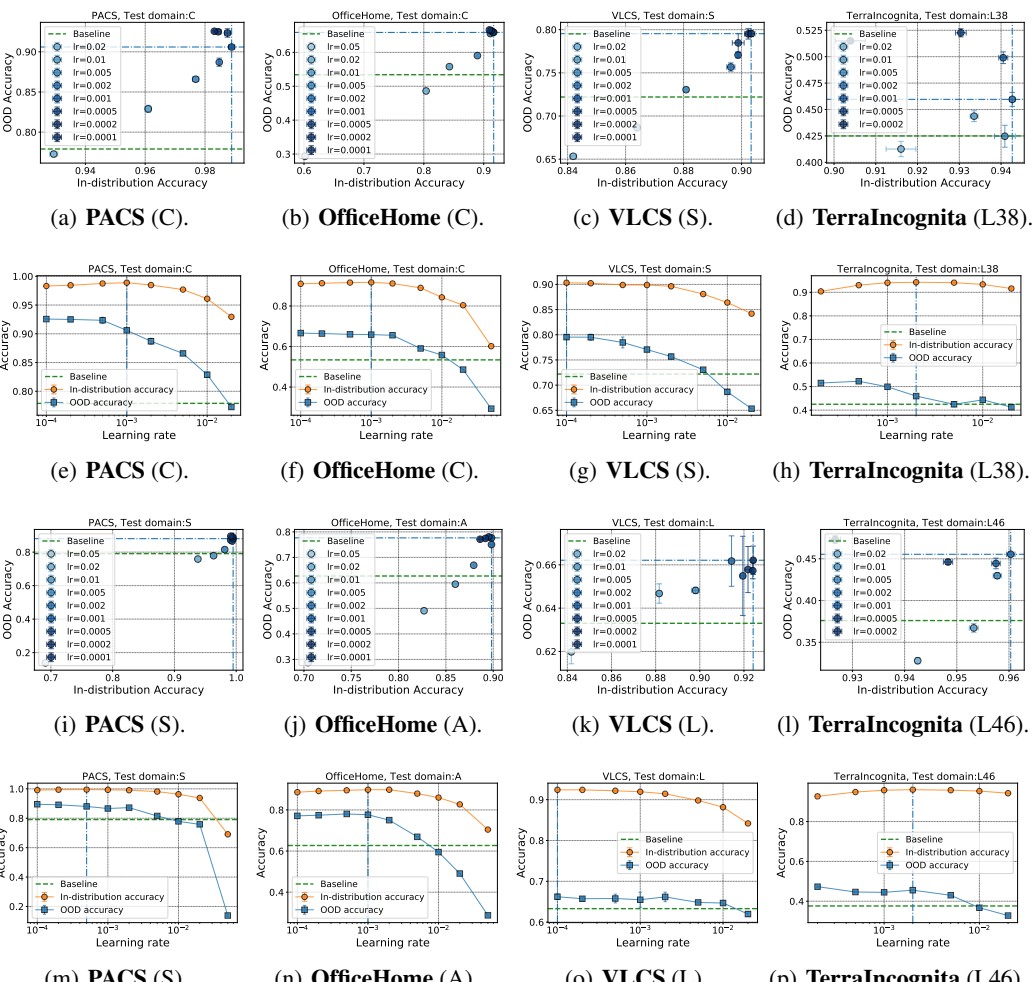

Figure 18: Evaluating models (SWSL-ResNext50-32x4d) fine-tuned by different learning rates (with *stage-wise learning rate decay*) on in-distribution and out-of-distribution data. (Top row) Scatter plot of in-distribution accuracy ($X$-axis) and out-of-distribution accuracy ($Y$-axis). (Bottom row) Compare in-distribution accuracy with out-of-distribution accuracy w.r.t. learning rate ($X$-axis). The green dashed line corresponds to the baseline OOD accuracy, and the blue dash-dot line represents the selected model (by selecting the model with best in-distribution accuracy).

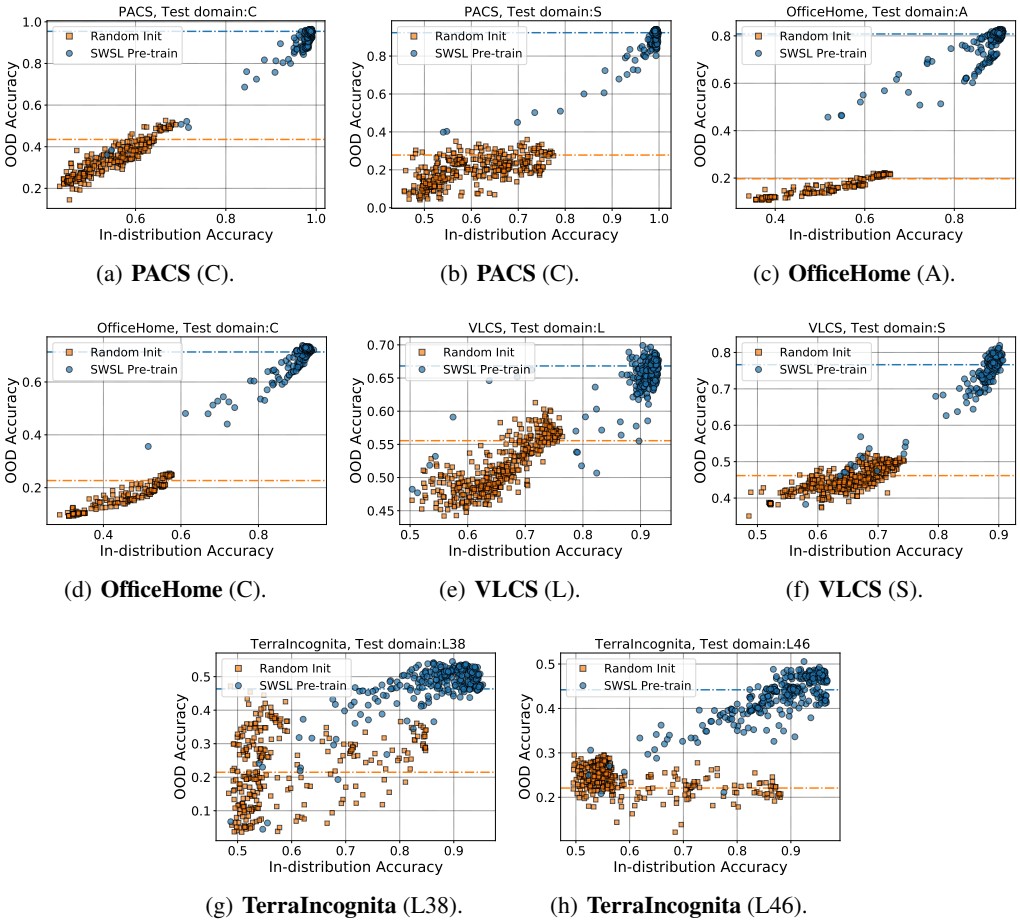

Figure 19: A comparison of ResNext101-32x8d models pre-trained on IG-1B-Targeted (SWSL-ResNext101-32x8d) v.s. ResNext101-32x8d without pre-training on out-of-distribution accuracy and in-distribution accuracy. The orange squares represent ResNext101-32x8d without pre-training and the blue circles represent models pre-trained on IG-1B-Targeted. We evaluate the models on both ID and OOD data every 100 iterations, and we visualize all the ID/OOD results in the above figures.

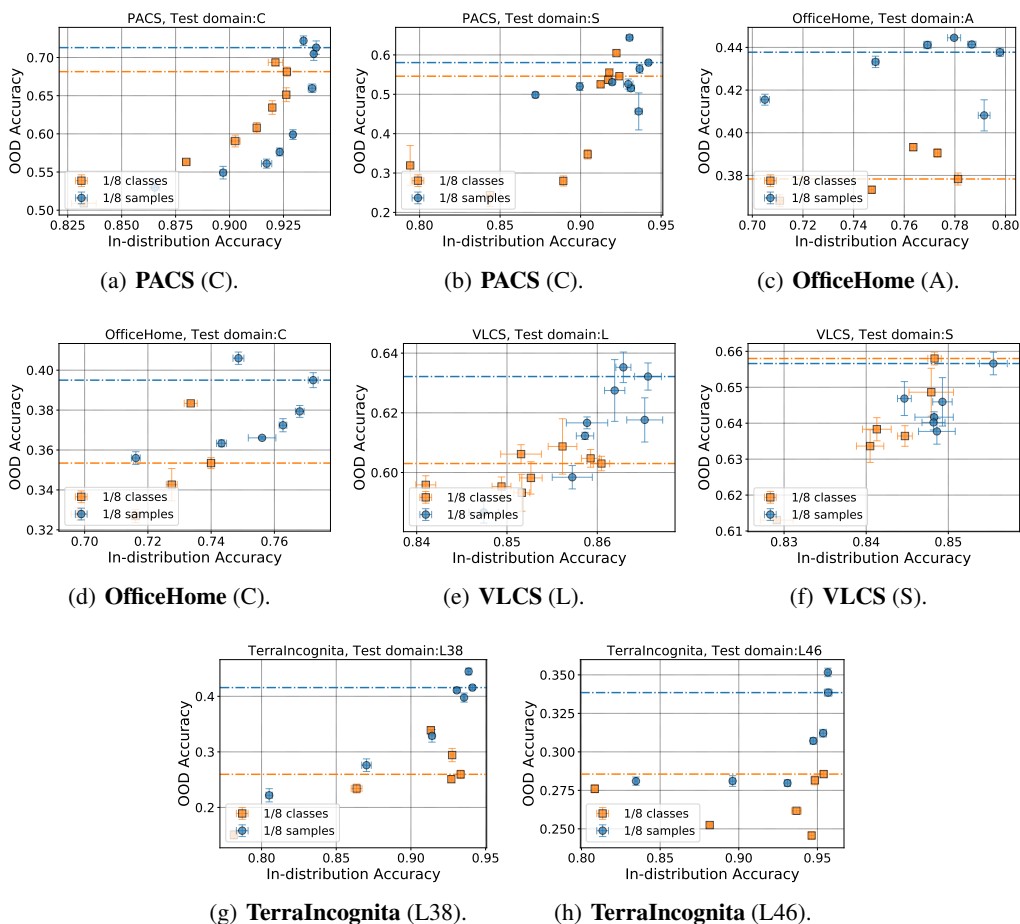

(a) **PACS** (C).

(b) **PACS** (C).

(c) **OfficeHome** (A).

(d) **OfficeHome** (C).

(e) **VLCS** (L).

(f) **VLCS** (S).

(g) **TerraIncognita** (L38).

(h) **TerraIncognita** (L46).

Figure 20: A comparison of ResNet50 pre-trained on 1/8 classes of the ImageNet-1k v.s. ResNet50 pre-trained on 1/8 training samples of the ImageNet-1k on out-of-distribution accuracy and in-distribution accuracy. The orange squares represent models pre-trained on 1/8 classes and the blue circles represent models pre-trained on 1/8 training samples.

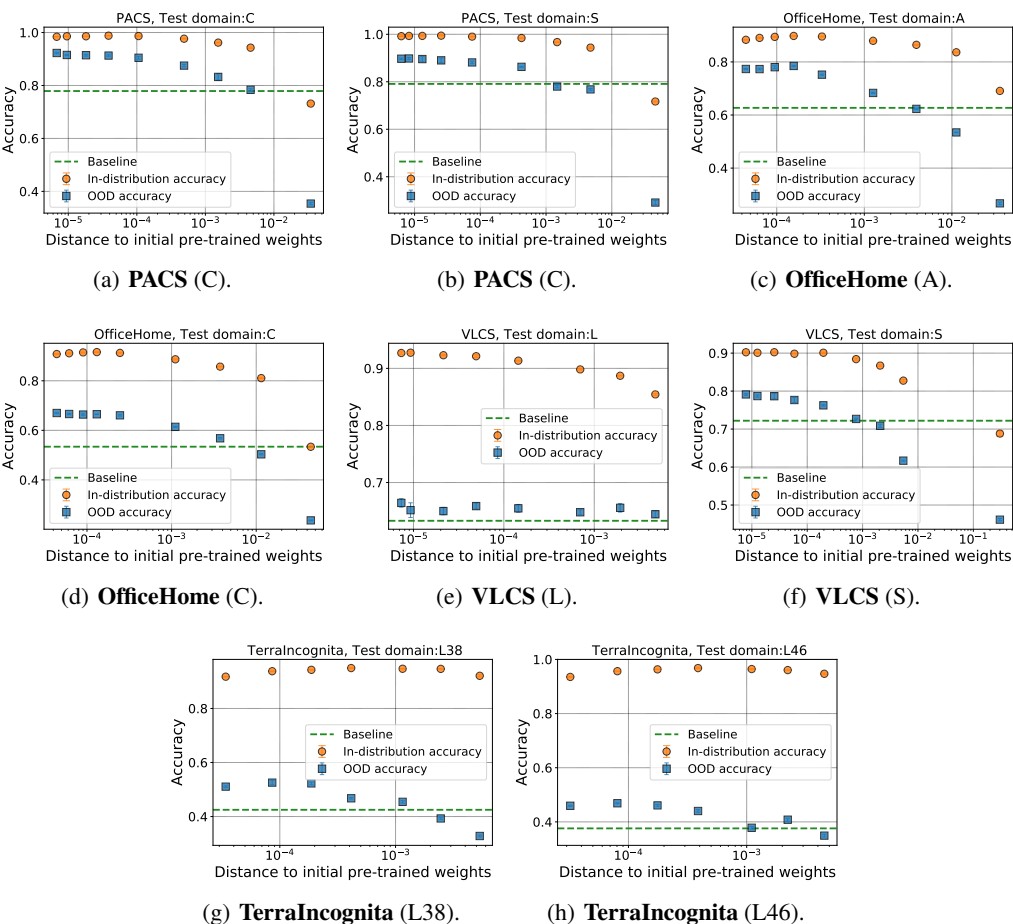

(a) **PACS** (C).

(b) **PACS** (C).

(c) **OfficeHome** (A).

(d) **OfficeHome** (C).

(e) **VLCS** (L).

(f) **VLCS** (S).

(g) **TerraIncognita** (L38).

(h) **TerraIncognita** (L46).

Figure 21: Evaluating models (SWSL-ResNext50-32x4d) fine-tuned by different learning rates on in-distribution and out-of-distribution data. $X$-axis represents the distance between the fine-tuned model weights and initial pre-trained model weights (i.e., $\|\theta_{\text{Fine-tuned}} - \theta_{\text{Init}}\|_2^2$). Each point in the scatter plot corresponds to one learning rate. The green dashed line corresponds to the baseline OOD accuracy. The orange squares represent the in-distribution accuracy results and the blue circles represent out-of-distribution accuracy results.

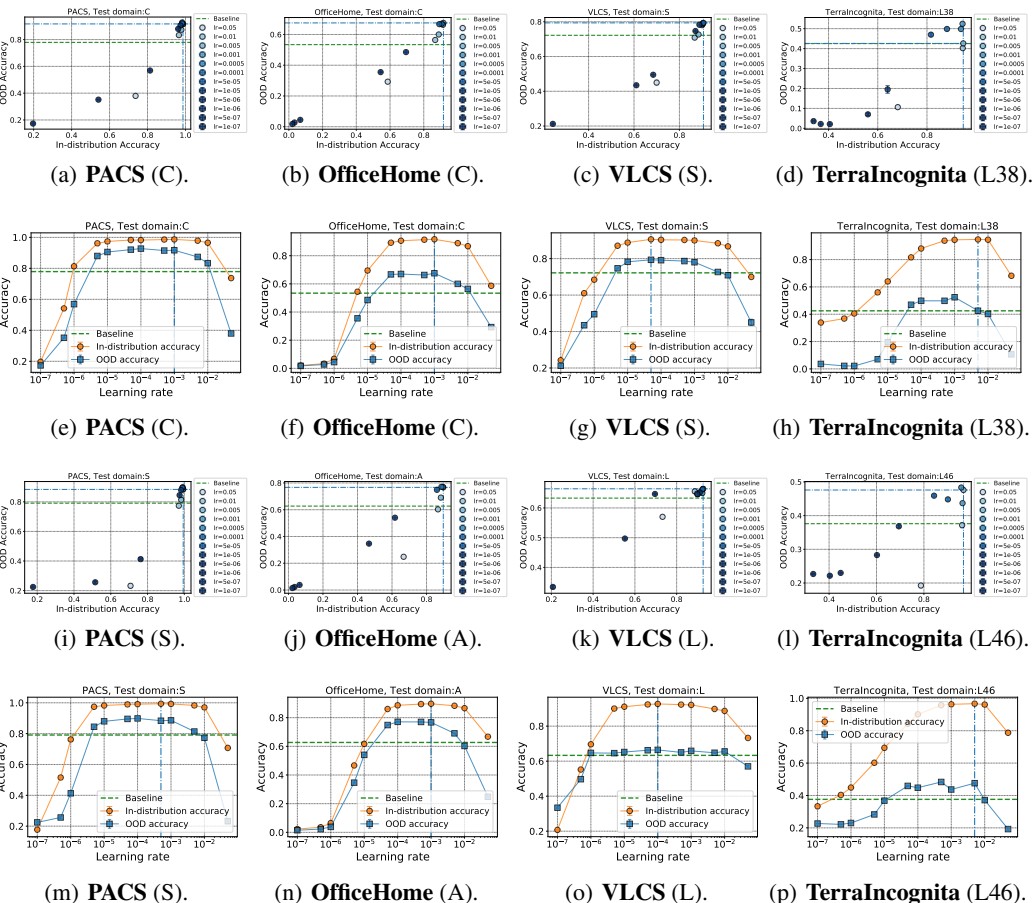

Figure 22: Evaluating models (SWSL-ResNext50-32x4d) fine-tuned by different learning rates (with *more learning rates, i.e.,* $\eta \in \{5 \times 10^{-2}, 1 \times 10^{-2}, 5 \times 10^{-3}, 1 \times 10^{-3}, 5 \times 10^{-4}, 1 \times 10^{-4}, 5 \times 10^{-5}, 1 \times 10^{-5}, 5 \times 10^{-6}, 1 \times 10^{-6}, 5 \times 10^{-7}, 1 \times 10^{-7}\}$) on in-distribution and out-of-distribution data. (Top row) Scatter plot of in-distribution accuracy ($X$-axis) and out-of-distribution accuracy ($Y$-axis). (Bottom row) Compare in-distribution accuracy with out-of-distribution accuracy w.r.t. learning rate ($X$-axis). The green dashed line corresponds to the baseline OOD accuracy, and the blue dash-dot line represents the selected model (by selecting the model with best in-distribution accuracy). *Here we visualize the results for all learning rates, and do not only show results for models that achieve* $> 95\%$ *training accuracy.*

