# OpenReview forum: "An Empirical Study of Pre-trained Models on Out-of-distribution Generalization"
_ICLR.cc/2022/Conference — ICLR 2022 Submitted_

### Official Review · Reviewer_jtZq · 2021-11-01

**Correctness:** 2
**Technical Novelty And Significance:** 1
**Empirical Novelty And Significance:** 2
**Recommendation:** 3
**Confidence:** 5

**Main Review:**

This paper provides extensive experimental results on four datasets to show how pre-trained data, pre-training model, learning rate and some ID tricks affect the OOD accuracy. Overall, the paper is easy to read and the organization is also good. The considered topic, OOD generalization, is interesting and important to the deep learning community. However, I have found some of the results unclear or incorrect.

1. The claim "models trained with smaller learning rates achieve much better OOD generalization" may be misleading/incorrect. It seems to me that the authors want to claim that the smaller lr, the better performance. This result is not well-supported. For example, Figure 2(h) shows that the performance with lr$=10^{-4}$ is worse than that with lr$=10^{-3}$; Figure 2 (e) and (f) show that the result for lr$=10^{-3}$ and the result for lr$=10^{-4}$ are almost same. Although the authors provide experiments with a range of learning rates between 0.05 and 0.0001, more experiments with smaller learning rates such as $10^{-5}$, $10^{-6}$ and $10^{-7}$ are needed.

2. The claim "methods listed in Table 2 does not significantly improve the OOD generalization across four datasets" is incorrect. We can see from Table 2 that both Label Smoothing (0.2) and SAM (0.02) are better than ERM (in Table 1) on 3 (of 4) datasets, while PatchGaussian (0.5) is better on 2 (of 4) datasets.

Thus, if above two issues cannot be addressed, then the main claims about learning rate and ID techniques are not correct/well-supported, which makes the contribution of this paper limited.

The novelty of using pre-training model to improve OOD generalization is not significant and this idea is not new. For example, see the ICML 2021 paper "Improved OOD Generalization via Adversarial Training and Pre-training". Although the authors show that the larger models and larger datasets need to be used simultaneously, but the contribution is incremental since the idea is too straightforward given the ICML paper. Furthermore, the intuition of using larger models and larger datasets simultaneously is unclear. It would be better if the authors could add some sentences to describe the intuition.

The contribution about ID tricks in this paper is mazy. It seems the paper's topic is about pre-training and fine-tuning on OOD generalization, which is not directly related to ID tricks. I am confused about adding ID tricks in this paper.

Two related papers are not discussed:

a. In-N-Out: Pre-Training and Self-Training using Auxiliary Information for Out-of-Distribution Robustness, ICLR 2021

b. Improved OOD Generalization via Adversarial Training and Pre-training, ICML 2021

**Summary Of The Paper:**

This paper empirically studies pre-trained model on OOD generalization. Specifically, the authors consider four datasets in this paper: PACS, Office-Home, VLCS and TerraIncognita. Based on the experimental results, the authors concludes three main observations . First,  larger models and larger datasets need to be simultaneously leveraged to improve OOD performance. Second, using smaller learning rates during fine-tuning is critical to achieving good results. Third, the strategies that improve in-distribution accuracy may lead to poor OOD performance.

**Summary Of The Review:**

1. The topic is important and interesting.

2. Two main contributions about learning rate and ID tricks may be incorrect or not well-supported.

3. The idea of using pre-training model is not new and the novelty of this paper is limited.

4. Unnecessary ID tricks and missed related work.

---

> ### Author Response · Authors · 2021-11-17
> **Reply to Reviewer jtZq [Part 1]**
>
> Thank you for your time and for your feedback!
>
> We would like to answer your questions/concerns below.
>
> ==========================================================================================
>
> **Q1**: The claim "models trained with smaller learning rates achieve much better OOD generalization" may be misleading/incorrect. This result is not well-supported. For example, Figure 2(h) shows that the performance with lr=10−4 is worse than that with lr=10−3; Figure 2 (e) and (f) show that the result for lr=10−3 and the result for lr=10−4 are almost same.
>
> **A1**: First of all, our claim is to describe an overall trend of the effect of the learning rate. Secondly, to better understand the effect of the learning rate for OOD generalization, we also need to study the corresponding in-distribution generalization. This is our motivation to provide the ID v.s. OOD scatter plot in Figure 2 (a)-(d). Our observation is, among models that achieve similar ID generalization performance, models trained with smaller learning rates will generalize better to OOD data. As shown in Figure 2 (a)-(d), the three examples listed above (Figure 2(h), (e), (f)) do not contradict our claim on the effect of the small learning rate. We also added a footnote on this claim in our revision (Page 5, footnote 1).
>
>
> **Q2**: Although the authors provide experiments with a range of learning rates between 0.05 and 0.0001, more experiments with smaller learning rates such as 10−5, 10−6 and 10−7 are needed.
>
> **A2**: Thank you for your suggestion. We had tried smaller learning rates in our experiments, and we found that too small learning rates usually cannot achieve >95% training accuracy. Therefore, we did not put the results in our initial submission. We have conducted experiments on $\eta \in [ 5\times 10^{-5}, 1\times 10^{-5}, 5\times 10^{-6}, 1\times 10^{-6}, 5\times10^{-7}, 1\times10^{-7} ] $ and summarized the results in Figure 22 in our revision (page 28). Here we visualize all the results without requiring ‘>95% training accuracy’. As we can see from Figure 22, among models that achieve reasonable in-distribution generalization, models fine-tuned with smaller learning rates will generalize better on OOD data. Overall, the updated results also align with our claim on the effect of smaller learning rates. We have updated these results in our revision.
>
>
> **Q3**: The claim "methods listed in Table 2 does not significantly improve the OOD generalization across four datasets" is incorrect.
>
> **A3**: Based on our results, we find that the four techniques (label smoothing, AutoAugment, PatchGaussian, and SAM) do not improve the OOD generalization significantly (< 3% improvement on OOD) compared with scaling model size and pre-training dataset size (5%-15% improvement on OOD). We have added ‘compared with scaling model size and pre-training dataset size’ in our revision on page 7.
>
>
> **Q4**: Thus, if above two issues cannot be addressed, then the main claims about learning rate and ID techniques are not correct/well-supported, which makes the contribution of this paper limited.
>
> **A4**: We hope our responses in **A1**, **A2**, **A3** could be helpful to clarify your concerns. We think the two claims are well-supported by extensive experiments in our submission.

---

> > ### Author Response · Authors · 2021-11-17
> > **Reply to Reviewer jtZq [Part 2]**
> >
> > **Q5**: The novelty of using pre-training model to improve OOD generalization is not significant and this idea is not new. For example, see the ICML 2021 paper "Improved OOD Generalization via Adversarial Training and Pre-training". Although the authors show that the larger models and larger datasets need to be used simultaneously, but the contribution is incremental since the idea is too straightforward given the ICML paper.
> >
> > **A5**: Thank you for the reference, and we have added this reference in our revision. [2] mainly studied how to leverage adversarial training to improve model generalization on corruption type OOD data (CIFAR10-C, ImageNet10-C), whereas we focus on the natural distributional shift. Meanwhile, [2] showed marginal improvement (<3% improvement) when applying adversarial pre-training (Table 3 in [2]), and they did not study the effectiveness of model size and pre-training dataset size for OOD generalization. In comparison, our work suggests that scaling both pre-training model size and dataset size can significantly improve OOD generalization.
> >
> >
> > **Q6**: The intuition of using larger models and larger datasets simultaneously is unclear. It would be better if the authors could add some sentences to describe the intuition.
> >
> > **A6**: As shown in Figure 6 (a), we identified that even without additional pre-training data, increasing model size can largely improve the OOD generalization. On the other hand, with the same model size, models pre-trained on more data generalize better on OOD. Our intuition is that scaling model size and pre-training dataset size simultaneously can be more effective for OOD generalization.
> >
> >
> > **Q7**: The contribution about ID tricks in this paper is mazy. It seems the paper's topic is about pre-training and fine-tuning on OOD generalization, which is not directly related to ID tricks. I am confused about adding ID tricks in this paper.
> >
> > **A7**: Our results in Section 3.1-3.3 suggest that simply fine-tuning better pre-trained models can significantly improve the OOD generalization, and our motivation to investigate the four techniques is to understand whether we could further push the OOD performance via these techniques. *Reviewer 1Vts* also appreciated the results of these techniques we studied in this paper. We believe these results on investigating training techniques can shed light on how to improve OOD generalization for future research.
> >
> > **Q8**: Two related papers [1, 2] are not discussed.
> >
> > **A8**: Thank you for your references, we have added and discussed these two references in our revision.
> >
> > ==========================================================================================
> >
> > Thank you again for your valuable feedback, and please let us know if you have any other concerns that we can address.
> >
> > [1] Sang Michael Xie, Ananya Kumar, Robbie Jones, Fereshte Khani, Tengyu Ma, Percy Liang, "In-N-Out: Pre-Training and Self-Training using Auxiliary Information for Out-of-Distribution Robustness", ICLR 2021.
> >
> > [2] Mingyang Yi, Lu Hou, Jiacheng Sun, Lifeng Shang, Xin Jiang, Qun Liu, Zhi-Ming Ma, "Improved OOD Generalization via Adversarial Training and Pre-training", ICML 2021.

---

### Official Review · Reviewer_5Ldh · 2021-11-01

**Correctness:** 3
**Technical Novelty And Significance:** 2
**Empirical Novelty And Significance:** 2
**Recommendation:** 5
**Confidence:** 3

**Main Review:**

On the methodology side, the paper is thorough, well-executed, with (mostly) clear results and well-designed experiments. Unfortunately, I think it lacks a strong selling point as most of the proposed insights are not novel enough.

For example, it is well-known that fine-tuning a pretrained model improves performance on downstream tasks, especially when those downstream tasks differ from upstream ones. See, for example, the work of Li et al. (2020) for an in-depth study on pretrained vision models, and Dodge et al. (2020) on language models. Similarly, the conclusion that larger models and datasets improve transfer is also well-studied — see the literature around "neural scaling laws", e.g., Kaplan et al. (2020) or Zhai et al. (2021) and references therein.

Maybe the most interesting contribution is the insight that smaller learning rates improve OOD accuracy even thought they perform on par with larger learning rates in ID data. But here the results are muddied, since this seems to be true for some settings but not all (p. 5, 1st paragraph). Could the authors expand on why ImageNet-pretrained models benefit from larger learning rates but not other models? What is their intuition for when practitioners should choose larger or smaller learning rates for OOD fine-tuning?

References:

- Hao Li, Pratik Chaudhari, Hao Yang, Michael Lam, Avinash Ravichandran, Rahul Bhotika, Stefano Soatto, "Rethinking the Hyperparameters for Fine-tuning", 2020.
- Jesse Dodge, Gabriel Ilharco, Roy Schwartz, Ali Farhadi, Hannaneh Hajishirzi, Noah Smith, "Fine-Tuning Pretrained Language Models: Weight Initializations, Data Orders, and Early Stopping", 2020.
- Jared Kaplan, Sam McCandlish, Tom Henighan, Tom B. Brown, Benjamin Chess, Rewon Child, Scott Gray, Alec Radford, Jeffrey Wu, Dario Amodei, "Scaling Laws for Neural Language Models", 2020.
- Xiaohua Zhai, Alexander Kolesnikov, Neil Houlsby, Lucas Beyer, "Scaling Vision Transformers", 2021.

**Summary Of The Paper:**

This paper empirically studies simple techniques to improve out-of-domain (OOD) generalization. First, they show that fine-tuning a pretrained model significantly improves OOD accuracy over the non-fine-tuned baseline. Then, they show that fine-tuning with smaller learning rates outperforms larger ones on OOD data, even though both get approximately identical in-domain (ID) accuracy. Finally, the conclude that larger models, larger datasets, or (preferably) both improve OOD generalization.

**Summary Of The Review:**

Although the experimental study of the paper is well executed, it provides few novel insights.

---

> ### Author Response · Authors · 2021-11-17
> **Reply to Reviewer 5Ldh**
>
> Thank you for your encouraging comments and valuable feedback!
>
> We would like to answer your questions/concerns below.
>
> ==========================================================================================
>
> **Q1**: It is well-known that fine-tuning a pre-trained model improves performance on downstream tasks, especially when those downstream tasks differ from upstream ones.
>
> **A1**: Yes, we agree that pre-training and fine-tuning can be very effective for improving performance on downstream tasks. Thank you for your references, we have added this reference in our revision.
> However, to the best of our knowledge, our work is the first to conduct a fine-grained analysis on the impact of pre-trained models for OOD generalization on vision benchmark datasets. In comparison, [1,2] studied fine-tuning on in-distribution generalization problems; [3] included results on the out-of-distribution performance of language models without any fine-tuning on NLP tasks, and [4] investigated the impact of vision transformer model size and pre-training data size on ImageNet-related datasets. None of the aforementioned work investigates the impact of pre-training & fine-tuning on the OOD generalization considered in our paper (i.e., fine-tuning on distribution P_{id} and generalizing to an unseen distribution P_{ood}, where both distributions share the same labeling function), and we believe our work provides concrete results and unique contributions for solving this problem.
>
> **Q2**: But here the results are muddied, since this seems to be true for some settings but not all (p. 5, 1st paragraph).
>
> **A2**: We agree that the smaller learning rates generalize better on OOD is not universally true across all settings. However, this phenomenon holds for most of the settings we considered in this paper, especially for models pre-trained on larger pre-training datasets than ImageNet-1k (including both SWSL models and BiT models). We have added ‘in most of the settings considered in this paper’ the description of the effect of learning rate (page 5, 1st paragraph) in our revision.
>
> **Q3**: Could the authors expand on why ImageNet-pretrained models benefit from larger learning rates but not other models? What is their intuition for when practitioners should choose larger or smaller learning rates for OOD fine-tuning?
>
> **A3**: The phenomenon ‘ImageNet-pretrained models benefit from larger learning rates’ mainly holds on one dataset (PACS), and smaller learning rates generalize better to OOD on other datasets in this paper. One possible explanation is that the ImageNet pre-trained dataset is not large enough to include similar samples as PACS, therefore, larger learning rates could lead to better generalization performance suggested by previous work [5] on the benefit of large learning rates. Regarding selecting learning rates for OOD fine-tuning, our work suggests that when the model is pre-trained on a relatively large dataset (e.g., ImageNet-21k and IG-1B-targeted), among the models that achieve similar in-distribution generalization performance, models trained with smaller learning rates will lead to better OOD generalization performance.
>
> ==========================================================================================
>
> Thank you again for your valuable feedback, and please let us know if you have any other concerns that we can address.
>
>
> [1]. Hao Li, Pratik Chaudhari, Hao Yang, Michael Lam, Avinash Ravichandran, Rahul Bhotika, Stefano Soatto, "Rethinking the Hyperparameters for Fine-tuning", 2020.
>
> [2]. Jesse Dodge, Gabriel Ilharco, Roy Schwartz, Ali Farhadi, Hannaneh Hajishirzi, Noah Smith, "Fine-Tuning Pretrained Language Models: Weight Initializations, Data Orders, and Early Stopping", 2020.
>
> [3]. Jared Kaplan, Sam McCandlish, Tom Henighan, Tom B. Brown, Benjamin Chess, Rewon Child, Scott Gray, Alec Radford, Jeffrey Wu, Dario Amodei, "Scaling Laws for Neural Language Models", 2020.
>
> [4]. Xiaohua Zhai, Alexander Kolesnikov, Neil Houlsby, Lucas Beyer, "Scaling Vision Transformers", 2021.
>
> [5]. Yuanzhi Li, Colin Wei, and Tengyu Ma. Towards explaining the regularization effect of initial large learning rate in training neural networks. arXiv preprint arXiv:1907.04595, 2019.

---

> > ### Comment · Reviewer_5Ldh · 2021-11-28
> > **Thanks for the response.**
> >
> > Thanks for taking the time to respond to my review.
> >
> > Since we know fine-tuning is important in other settings, the provided insights on OOD might be novel but not significant enough for ICLR.
> > Moreover, the insights are not incisive enough as shown in your answers A2 and A3 -- for example, if the issue is that ImageNet is "not large enough to contain samples similar samples as PACS", doesn't that further undermine the claims on the other settings which wouldn't be different enough to be considered OOD?
> >
> > Overall I'm still tending towards rejection for now, since those insights need to be more carefully thought out.

---

> > > ### Author Response · Authors · 2021-11-29
> > > **Re: Thanks for the response**
> > >
> > > Thank you for your response and for recognizing the novelty of our work!
> > >
> > > ==========================================================================================
> > >
> > > **Q1:** if the issue is that ImageNet is "not large enough to contain samples similar samples as PACS", doesn't that further undermine the claims on the other settings which wouldn't be different enough to be considered OOD?
> > >
> > > **A1:** The '*One possible explanation is that the ImageNet pre-trained dataset is not large enough to include similar samples as PACS*' is served as one possible intuition on the effect of small learning rates on ImageNet, which does not undermine our claims in this paper.
> > >
> > > First of all, models pre-trained on all three pre-training datasets (ImageNet/ImageNet-21k/IG-1B) have been widely adopted in the OOD generalization literature, it is important to have a better/systematic understanding of the role of the pre-trained models in the pre-train/fine-tune framework as studied in our work. Meanwhile, our work identifies (a). the effectiveness of scaling model size (refer to Figure 6.(a), larger model pre-trained on ImageNet-1k improves OOD generalization on PACS); (b). learning rates play an important role in OOD generalization across different pre-trained models (Figure 12, 13), smaller learning rates will lead to better OOD generalization in most cases. Both the above arguments hold on ImageNet, thus we do not think our contributions are undermined. Secondly, we believe our findings on the effect of small learning rates, through our carefully designed and extensive experiments, can provide valuable insights for future research on improving OOD generalization.
> > >
> > > ==========================================================================================
> > >
> > > We hope our response clarifies your concern. We are happy to answer any remaining questions/concerns.

---

### Official Review · Reviewer_mKe9 · 2021-11-02

**Correctness:** 3
**Technical Novelty And Significance:** 1
**Empirical Novelty And Significance:** 2
**Recommendation:** 5
**Confidence:** 2

**Main Review:**

I will preface by saying that I am not intimately familiar with the computer-vision literature and since, despite its unnecessarily general title and abstract, this paper is very computer-vision specific, I may be overlooking some finer details.

That being said, I think that the paper places itself nicely within the existing literature by casting the focus on OOD generalization specifically. While the conclusions (that larger models and more diverse datasets are better) are not unexpected, such detailed empirical studies are worthwhile for practitioners and researchers alike. However I think the paper falls short by not going one step further and providing additional insight to try and explain some of its (many) experimental findings. This makes it harder to tell whether and how these results would transfer to other architectures, modalities or datasets considered here, which ultimately limits the impact of the paper. I think the paper is in a good shape, but it would need additional ablations and analysis to warrant acceptance at ICLR.

Strengths:
- While the overall conclusion that larger models pre-trained on more (and more diverse) are better is not surprising, I think it is useful to confirm this finding finding with respect to OOD generalization specifically.
- Detailed analysis and experimental results, which should be useful for practitioners in the field
- Experimental results are overall well presented

Weaknesses:
- Unnecessarily general title and abstract: the paper is very much vision specific and this should be made clear from the get go. As it stands I found the opening of the paper rather misleading. There is some work outside of vision which looked at this particular problem and bears mentioning, for instance in NLP Hendricks et al. "Pretrained Transformers Improve Out-of-Distribution Robustness" https://arxiv.org/pdf/2004.06100.pdf
- While there are a lot of experimental results, I think the paper suffers from not going one step further and investigating the underlying causes behind a lot of the results. As it stands, the reader is left with rather vague and somewhat expected conclusions: "larger and more diverse datasets are better" and "smaller learning rates are better".
   * What specifically in the larger datasets helps? Is it really diversity or size? For example, if IG-1B-Targeted is downsized to the size of Imagenet, do the improvements observed persist?
   * What about the smaller learning rates make them more suitable? I have a suspicion that it has to do with the weights deviating less from their pre-trained starting point during fine-tuning. It would be interested to look at the interaction between "catastrophic forgetting" of the pre-training task and OOD generalization.

**Summary Of The Paper:**

Within the now common pre-train/fine-tune training paradigm, this paper provide an empirical examination of the influence of pre-training on out-of-distribution accuracy *after* fine-tuning, specifically for computer vision. The authors showcase evidence suggesting that the choice of the pre-training dataset, the size of the model and the fine-tuning learning rate directly influence downstream OOD accuracy throughout a series of tasks. In addition, they provide some analysis of the effect that various training techniques (eg. label smoothing, augmentation etc..) have on OOD accuracy when they are used at fine-tuning time.

**Summary Of The Review:**

I think that the paper places itself nicely within the existing literature by casting the focus on OOD generalization specifically. While the conclusions (that larger models and more diverse datasets are better) are not unexpected, such detailed empirical studies are worthwhile for practitioners and researchers alike. However I think the paper falls short by not going one step further and providing additional insight to try and explain some of its (many) experimental findings. This makes it harder to tell whether and how these results would transfer to other architectures, modalities or datasets considered here, which ultimately limits the impact of the paper. I think the paper is in a good shape, but it would need additional ablations and analysis to warrant acceptance at ICLR.

---

> ### Author Response · Authors · 2021-11-17
> **Reply to Reviewer mKe9**
>
> Thank you for your encouraging comments and valuable feedback! We would like to answer your questions/concerns below.
>
> ==========================================================================================
>
> **Q1**: Unnecessarily general title and abstract.
>
> **A1**: Thank you for your suggestion on the title and abstract. In our revision, we have changed our title as ‘An Empirical Study of Pre-trained Vision Models on Out-of-distribution Generalization’ and highlighted ‘image classification’ in our abstract.
>
>
> **Q2**: What specifically in the larger datasets helps? Is it really diversity or size? For example, if IG-1B-Targeted is downsized to the size of Imagenet, do the improvements observed persist?
>
> **A2**: Thank you for suggesting the insightful experiments! Due to limited time, we conducted experiments on performing pre-training on ImageNet-1k to investigate the impact of the diversity of the pre-training data size. We consider two pre-training datasets with the same number of training samples, (1). We randomly select a 125 class subset of ImageNet-1k (which has 1,000 classes), i.e. this pre-training dataset has 1/8 samples of the original dataset, denoted by [1/8 classes]; (2). We randomly subsample the ImageNet-1k by a factor of 8, i.e., this pre-training dataset also has 1/8 samples of the original dataset, denoted by [1/8 samples]. Although both datasets have similar training samples, [1/8 samples] is supposed to be more diverse than [1/8 classes] since it contains more classes.
>
> Next we pre-train models (with the same architecture ResNet50) on these two pre-training datasets separately, and then fine-tune two ResNet50s on the training datasets we considered in this paper. We have summarized the results in Figure 20 in our revision (page 26). We can find that models pre-trained on the more diverse dataset perform much better on OOD data in most cases we considered in our paper. *We believe this set of experiments provide more evidence on the importance of the diversity of the pre-training dataset*. We have updated these results in our revision.
>
> **Q3**: I have a suspicion that it has to do with the weights deviating less from their pre-trained starting point during fine-tuning. It would be interested to look at the interaction between "catastrophic forgetting" of the pre-training task and OOD generalization.
>
> **A3**: Thank you for suggesting the interesting experiments. We have conducted experiments to investigate the interaction between the norm of the weights difference and the OOD generalization. We have conducted new experiments based on your suggestion. More specifically, for each model fine-tuned by one learning rate, we compute the distance between the fine-tuned model weights and the initial model weights, i.e., the distance metric is defined $||\theta_{\text{Fine-tuned}} - \theta_{\text{Init}}||_{2}^{2}$, where $\theta$ represents the model weights. As shown in Figure 21 (page 27), we can observe that there is a clear relation between the distance to the initial model and the ID/OOD generalization of the fine-tuned models. We believe this set of new experiments further justifies the impact of the regularization effect of small learning rates we have identified in this paper. Thank you again for your insightful suggestion. We have updated these results in our revision.
>
>
> **Q4**: This makes it harder to tell whether and how these results would transfer to other architectures, modalities or datasets considered here, which ultimately limits the impact of the paper.
>
> **A4**: We would like to highlight that our work includes a large set of pre-trained models (including different architectures and different pre-trained datasets) and the commonly used domain generalization datasets. Given the OOD generalization in computer vision is an important problem, we believe our fine-grained analysis on the impact of pre-trained models for OOD generalization will provide both practical guidance for practitioners and empirical insights for future theoretical study.
>
> ==========================================================================================
>
> Thank you again for your valuable feedback, and please let us know if you have any other concerns that we can address.

---

### Official Review · Reviewer_1Vts · 2021-11-05

**Correctness:** 3
**Technical Novelty And Significance:** 2
**Empirical Novelty And Significance:** 3
**Recommendation:** 6
**Confidence:** 3

**Main Review:**

Overall, I think this paper provides a good set of empirical studies on an important problem: what contributes to the better OOD generalization performance of pre-trained models (and relatedly, how are ID accuracies related to these OOD accuracies).

Strengths:
1. Extensive experiments. I especially like that the authors tried to tackle at this problem from multiple angles, such as LR, model size, model type, augmentations, etc.
2. The paper is overall well-written and easy to understand.
3. The underlying problem studied here is very important and while some of the insights have been brought up by prior works, this paper provides an in-depth analysis of how pre-trained models behave (under the influence of all these different factors) when there's a distribution shift. I have no doubt that the conclusion of this paper will be very useful for future work.
4. Good results and on large scales. The authors are able to take advantage of all these findings to produce good OOD generalization outcomes.

Weaknesses/problems:
1. The usefulness of pre-training was already studied before (e.g., [1,2]), though in somewhat different directions. This paper puts more emphasis on training mechanisms, which provide useful, interesting, but still incremental insights.
2. The conclusions are still empirical, domain-specific, and have some important questions (in my mind) unanswered. For example, since learning rate was a major factor studied in the paper, it makes sense to consider the effect of the learning rate annealing schedule as well, which is a technique very commonly used in training image classification models. As another example, the paper mentions multiple times how a **"more diverse"** dataset for pretraining would help (e.g., IG-1B-Targeted vs. ImageNet-1k). However, I feel that this is a more relative and subjective measure--- e.g., how do we know a'priori whether a dataset is "diverse enough"?
3. I like the discussion on ID and OOD accuracy, so a related question: let's say we train 1) a model A on dataset C to reach 70% accuracy on C's test set; and 2) a model B (with the same **architecture** as A) on a larger pretraining dataset D first, and then fine-tune a bit (just a bit, not thoroughly; for example, you can early stop) on C to make it also reach 70% accuracy on the downstream dataset C's test set. Does the conclusion of this paper imply model B will always have better OOD performance than model A? This is different from the results presented in Table 8 because the models there have substantially worse ID accuracy as well.



[1] http://proceedings.mlr.press/v139/yi21a/yi21a.pdf
[2] https://aclanthology.org/2020.acl-main.244.pdf

**Summary Of The Paper:**

This paper provides a thorough and relatively in-depth analysis of how pre-trained models affect OOD generalization performance. Specifically, the authors dissect this issue and perform extensive ablations on learning rates, model architecture, dataset size, model size, etc. By studying large-scale pretraining on vision tasks, the paper claims that 1) pre-trained models typically have better OOD generalization performance (which prior works have somewhat discussed as well); 2) small learning rates during the fine-tuning phase tend to create more robust models; and 3) some of the previous understandings (e.g., of how ID and OOD accuracies are related; or of the effect of training data size) might need to be revisited.

**Summary Of The Review:**

I recommend (weak) acceptance of this paper because 1) the conclusions drawn in this paper, while somewhat incremental, are still very useful and would be a great addition to this line of work; and 2) the experiments are thorough and realistic. I don't identify any obvious flaw in the methodology or analysis.

---------------------

Post-rebuttal: See my comment below. I'm updating the score to 7 (which doesn't exist in the ICLR review form, but just to show that I'm advocating more for acceptance).

---

> ### Author Response · Authors · 2021-11-17
> **Reply to Reviewer 1Vts**
>
> Thank you for your encouraging comments and valuable feedback! Thank you for your reference [1], we have added this reference in our revision.
>
> We would like to answer your questions/concerns below.
>
> ==========================================================================================
>
> **Q1**: Since learning rate was a major factor studied in the paper, it makes sense to consider the effect of the learning rate annealing schedule as well.
>
> **A1**: We applied ​​the cosine learning rate decay in our experiments (as mentioned in the Fine-tuning and model selection paragraph in Section 2). We have also tried another learning decay strategy (stage-wise learning rate decay, i.e., decay learning rate by a factor of 10 every 2000 iterations) and the overall phenomenon is similar to the cosine learning rate decay (as shown in Figure 18 in our revision, page 24). We have updated these results in our revision.
>
>
> **Q2**: How do we know a'priori whether a dataset is "diverse enough"?
>
> **A2**: In general, it is nontrivial to estimate the diversity of the pre-training dataset. One useful approach is to examine the performance of pre-trained vision models on a diverse set of out-of-distribution benchmark datasets: ImageNet-V2, ImageNet-C, ImageNet-R, ImageNet-A, etc. The comprehensive evaluation will be helpful to determine whether the pre-trained model is robust to a diverse set of OOD data, and provide more intuitions on whether the pre-trained dataset is diverse enough for handling different types of OOD data. On the other hand, the set of pre-trained models (SWSL), which achieve the best OOD performance in our work, also outperforms other pre-trained models on the above-mentioned OOD benchmark datasets in most cases.
>
> **Q3**: I like the discussion on ID and OOD accuracy, so a related question: let's say we train 1) a model A on dataset C to reach 70% accuracy on C's test set; and 2) a model B (with the same architecture as A) on a larger pretraining dataset D first, and then fine-tune a bit (just a bit, not thoroughly; for example, you can early stop) on C to make it also reach 70% accuracy on the downstream dataset C's test set. Does the conclusion of this paper imply model B will always have better OOD performance than model A? This is different from the results presented in Table 8 because the models there have substantially worse ID accuracy as well.
>
> **A3**: Thank you for your insightful suggestion. We have conducted experiments to compare the randomly initialized model v.s. the pre-trained model with the same architecture on the four datasets we considered in the paper. For each model with every learning rate chosen, we evaluate the ID accuracy and OOD accuracy every 100 iterations and plot all the evaluations in scatter plots. As shown in Figure 19 in our revision (page 25), with similar ID accuracy, the pre-trained models significantly outperform random initialized models w.r.t. OOD accuracy on some datasets. We have updated these results in our revision.
>
> ==========================================================================================
>
>
> Thank you again for your valuable feedback, and please let us know if you have any other concerns that we can address.
>
> [1] Mingyang Yi, Lu Hou, Jiacheng Sun, Lifeng Shang, Xin Jiang, Qun Liu, Zhi-Ming Ma, "Improved OOD Generalization via Adversarial Training and Pre-training", ICML 2021.

---

> > ### Comment · Reviewer_1Vts · 2021-11-30
> > **Thank you for the response!**
> >
> > I have read the reviews and the authors' responses. I'd like to thank the authors for the new experiment results added (especially the Q3 here; interesting results). However, as almost all reviewers pointed out, this is a good empirical study but there are some remaining questions that surely needs to be answered (on learning rate, for example). I'm upgrading my score to 7 (not a score that exists in ICLR this year but I'll let AC know), while strongly encouraging the authors to add more comprehensive empirical analysis to the rebuttal.

---

> > > ### Author Response · Authors · 2021-11-30
> > > **Thanks for the response from Reviewer 1Vts**
> > >
> > > We genuinely thank you for the constructive suggestions to make our work better! We will include more empirical analysis based on other reviewers' feedback and update those results in our camera-ready version.

---

### Author Response · Authors · 2021-11-27
**Looking for feedback from the reviewers: happy to clarify any remaining questions**

Thanks for your hard work reviewing! We are looking for feedback on whether the points made in the reviews have now been addressed.

About the revised version: We have modified our paper based on the suggestions/concerns of the reviewers and have uploaded a revised version. All revised places are marked in blue color. In particular, we have made the following main updates:

**1.** For reviewer *1Vts*, we have (a). provided new experiments on another learning decay strategy (Figure 18, page 24) in our revision; (b). updated detailed comparisons between pre-trained models and models without pre-training (Figure 19, page 25) in our revision.

**2.** For reviewer *mKe9*, we have (a). provided new experimental results on investigating the effect of data diversity under the same number of training samples (Figure 20, page 26) in our revision; (b). conducted new experiments on exploring the interaction between catastrophic forgetting and OOD generalization (Figure 21, page 27) in our revision.

**3.** For reviewer *5Ldh*, we have improved our presentation of the first paragraph on page 5 and added related references in our revision.

**4.** For reviewer *jtZq*, we have (a). included new experiments on smaller learning rates (Figure 22, page 28) in our revision; (b). improved our presentation of the training techniques on page 7 in our revision.


The updated experimental results align well with our previous results and further strengthen our findings OOD generalization. We hope the revision has addressed all your concerns. We are happy to answer any remaining questions regarding our rebuttal or the paper itself.

---

### Decision · Program_Chairs · 2022-01-20

**Decision:**

Reject

**Comment:**

The paper presented an empirical study of pre-trained models on the Out-of-distribution Generalization problem.
Authors evaluated various factors (such as model sizes, datasets, learning rate, etc) and claim some major findings:  1) larger models have better OOD generalization, and combining both larger models and larger datasets is critical; 2) smaller learning rate during fine-tuning is critical; 3) strategies improving in-distribution accuracy may hurt OOD. Overall, this paper is a well-written empirical study with some useful insights, but the new findings from the empirical studies are generally not surprising and the overall contribution is not significant enough for acceptance.